# Efficient and Differentiable Conformal Prediction with General Function Classes

**Yu Bai**
Salesforce Research
yu.bai@salesforce.com

**Song Mei**
UC Berkeley
songmei@berkeley.edu

**Huan Wang & Yingbo Zhou & Caiming Xiong**
Salesforce Research
{huan.wang, yingbo.zhou, cxiong}@salesforce.com

## Abstract

Quantifying the data uncertainty in learning tasks is often done by learning a prediction interval or prediction set of the label given the input. Two commonly desired properties for learned prediction sets are *valid coverage* and *good efficiency* (such as low length or low cardinality). Conformal prediction is a powerful technique for learning prediction sets with valid coverage, yet by default its conformalization step only learns a single parameter, and does not optimize the efficiency over more expressive function classes.

In this paper, we propose a generalization of conformal prediction to multiple learnable parameters, by considering the constrained empirical risk minimization (ERM) problem of finding the most efficient prediction set subject to valid empirical coverage. This meta-algorithm generalizes existing conformal prediction algorithms, and we show that it achieves approximate valid population coverage and near-optimal efficiency within class, whenever the function class in the conformalization step is low-capacity in a certain sense. Next, this ERM problem is challenging to optimize as it involves a non-differentiable coverage constraint. We develop a gradient-based algorithm for it by approximating the original constrained ERM using differentiable surrogate losses and Lagrangians. Experiments show that our algorithm is able to learn valid prediction sets and improve the efficiency significantly over existing approaches in several applications such as prediction intervals with improved length, minimum-volume prediction sets for multi-output regression, and label prediction sets for image classification.

## 1 Introduction

Modern machine learning models can yield highly accurate predictions in many applications. As these predictions are often used in critical decision making, it is increasingly important to accompany them with an uncertainty quantification of how much the true label may deviate from the prediction. A common approach to quantifying the uncertainty in the data is to learn a *prediction set*—a set-valued analogue of usual (point) predictions—which outputs a *subset* of candidate labels instead of a single predicted label. For example, this could be a prediction interval for regression, or a discrete label set for multi-class classification. A common requirement for learned prediction sets is that it should achieve *valid coverage*, i.e. the set should cover the true label with high probability (such as 90%) on a new test example (Lawless & Fredette, 2005). In addition to coverage, the prediction set is often desired to have a good *efficiency*, such as a low length or small cardinality (Lei et al., 2018; Sadinle et al., 2019), in order for it to be informative. Note that coverage and efficiency typically come as a trade-off, as it is in general more likely to achieve a better coverage using a larger set.

This paper is concerned with the problem of finding the most efficient prediction set with valid coverage. Our approach builds on conformal prediction (Vovk et al., 2005), a powerful framework for generating prediction sets from (trained) base predictors with finite-sample coverage guarantees.

Code available at https://github.com/allenbai01/cp-gen.

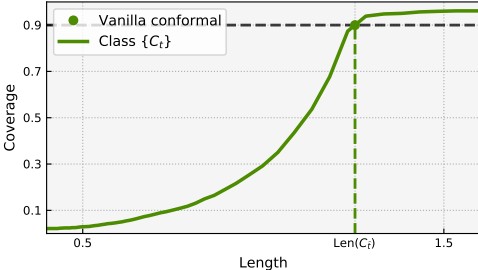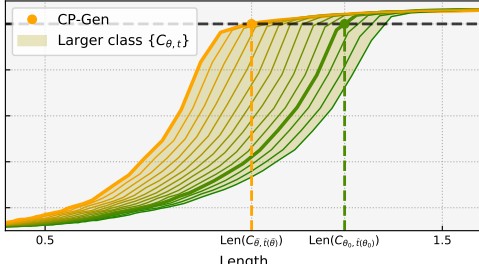

Figure 1: Comparison of vanilla conformal prediction and our `CP-Gen` for learning a prediction interval with 90% nominal coverage on a real-world regression task. Left: The function class $\{C_t\}$ is the prediction intervals used by Conformalized Quantile Regression. Right: $\{C_{\theta,t}\}$ is a larger class of intervals used by our conformal quantile finetuning procedure with the same base predictor; here the additional trainable parameter $\theta$ is the last linear layer within the quantile neural network (cf. Section 5.1 and Appendix E.3 for more details).

Conformal prediction has been used for learning prediction sets in a variety of tasks in regression (Lei & Wasserman, 2014; Lei et al., 2018; Romano et al., 2019), classification (Cauchois et al., 2020b; Romano et al., 2020; Angelopoulos et al., 2020), structured prediction (Bates et al., 2021), and so on. However, the conformalization step in conformal prediction by default does not offer the flexibility for optimizing additional efficiency metrics, as the efficiency is already determined by the associated score function and the target coverage level. As a concrete example, the Conformalized Quantile Regression algorithm learns a single width adjustment parameter that turns a two-sided quantile predictor into a prediction interval of valid coverage (Romano et al., 2019); however, it does not offer a way of further optimizing its length (cf. Figure 1 Left).

For certain efficiency metrics and prediction tasks, several approaches have been proposed, for example by designing a better score function (Angelopoulos et al., 2020), using base predictors of a specific form (Izbicki et al., 2019; 2020; Sadinle et al., 2019), or selecting a best training hyperparameter (Yang & Kuchibhotla, 2021). However, optimizing the efficiency for more general tasks or efficiency metrics still largely requires "manual" efforts by the researcher, as it (1) often relies on specific domain knowledge about the task at hand; (2) is often done in conjunction with conformal prediction in multiple rounds of trial-and-error; (3) is often done by reasoning about high-level properties of the efficiency loss and coverage constraints (e.g. what makes the length short), but not by directly optimizing the efficiency-coverage trade-off in a data-dependent way. To the best of our knowledge, there is a lack of a more principled and unified approach for optimizing any efficiency metric subject to valid coverage over any class of prediction sets.

In this paper, we cast the above task as a constrained empirical risk minimization (ERM) problem of optimizing the efficiency subject to the coverage constraint, over any general function class of prediction sets with potentially multiple learnable parameters. This is motivated by a simple observation that vanilla conformal prediction is already equivalent to solving such a constrained ERM with one learnable parameter (Section 2.1). Overall, our algorithm can be viewed as an automatic and data-dependent approach for optimizing the efficiency simulatneously with conformal prediction. Our contributions are summarized as follows.

- We propose `CP-Gen` (Conformal Prediction with General Function Class), a generalization of conformal prediction to learning multiple parameters. `CP-Gen` selects within an arbitrary class of prediction sets by solving the constrained ERM problem of best efficiency subject to valid empirical coverage (Section 3.1), and is a systematic extension of existing algorithms.

- We show theoretically that `CP-Gen` achieves approximately valid coverage and near-optimal efficiency within class, whenever the class is low-capacity with respect to both the coverage and the efficiency loss (Section 3.2, with concrete examples in Appendix C). We also provide a practical variant `CP-Gen-Recal` using data splitting and reconformalization, which achieves exact coverage, as well as good efficiency under additional assumptions (Section 3.3).

- To address the issue that `CP-Gen` and `CP-Gen-Recal` involve a non-differentiable coverage constraint, we develop a differentiable approximation using surrogate losses and Lagrangians (Section 4). This allows us to solve the constrained ERM problem over higher-dimensional continuous parameter spaces via gradient-based optimization, and is more flexible than existing algorithms that require discretization and brute-force search.

- We empirically demonstrate that `CP-Gen-Recal` with our gradient-based implementation can learn prediction sets with valid coverage and significantly improved efficiency on three real-data tasks: prediction intervals for regression with improved length, minimum-volume prediction sets for multi-output regression, and label prediction sets for ImageNet (Section 5 & Appendix F).

We illustrate our main insight via the coverage-vs-efficiency trade-off plots in Figure 1: While vanilla conformal prediction only learns a single parameter (within its conformalization step) by a simple thresholding rule over a coverage-efficiency *curve*, our `CP-Gen` is able to further improve the efficiency by thresholding a *region* formed by a larger function class.

## 1.1 RELATED WORK

**Learning prediction sets via conformal prediction**    The framework of conformal prediction for learning prediction sets is originated in the early works of (Vovk et al., 1999; 2005; Shafer & Vovk, 2008). The main advantage of conformal prediction is that it yields (marginal) coverage guarantees regardless of the data distribution (i.e. distribution-free). More recently, conformal prediction has been applied to a variety of uncertainty quantification tasks, such as prediction intervals for regression (Papadopoulos, 2008; Vovk, 2012; 2015; Lei & Wasserman, 2014; Vovk et al., 2018; Lei et al., 2018; Romano et al., 2019; Izbicki et al., 2019; Guan, 2019; Gupta et al., 2019; Kivaranovic et al., 2020; Barber et al., 2021; Foygel Barber et al., 2021), label prediction sets for classification problems (Lei et al., 2013; Sadinle et al., 2019; Romano et al., 2020; Cauchois et al., 2020b;a; Angelopoulos et al., 2020), and prediction sets for structured output (Bates et al., 2021).

**Optimizing efficiency in addition to valid coverage**    The problem of finding a prediction set with (approximate) valid coverage and small size has been considered, e.g. in Pearce et al. (2018); Chen et al. (2021) for regression and Park et al. (2019) for classification; however, these approaches do not use conformal prediction. Yang & Kuchibhotla (2021) propose to minimize the length of the conformal interval over either a finite class or a linear aggregation of base predictors, and provides coverage and efficiency guarantees. All above works formulate this task as a risk minimization problem, yet are restricted to considering either finite classes or specific efficiency loss functions. Our work is inspired by (Yang & Kuchibhotla, 2021) and generalizes the above works by allowing any function class and efficiency loss, along with providing a differentiable approximate implementation.

The problem of optimizing the efficiency can also be done by utilizing structures of the particular efficiency loss to choose a specific base predictor and an associated prediction set (Lei & Wasserman, 2014; Sadinle et al., 2019; Izbicki et al., 2019; 2020). By contrast, our approach does not require either the efficiency loss or the base predictor to possess any structure, and is thus complementary.

**Other algorithms and theory**    An alternative line of work constructs prediction intervals / prediction sets by aggregating the prediction over multiple base predictors through Bayesian neural network (Mackay, 1992; Gal & Ghahramani, 2016; Kendall & Gal, 2017; Malinin & Gales, 2018; Maddox et al., 2019) or ensemble methods (Lakshminarayanan et al., 2016; Ovadia et al., 2019; Huang et al., 2017; Malinin et al., 2019). However, these methods do not typically come with (frequentist) coverage guarantees. The recent work of Hoff (2021) studies ways of enhancing Bayes-optimal prediction with frequentist coverage. Prediction intervals can also be obtained by parameter estimation using a parametric model for the data (Cox, 1975; Bjornstad, 1990; Beran, 1990; Barndorff-Nielsen & Cox, 1996; Hall et al., 1999; Lawless & Fredette, 2005); see (Tian et al., 2020) for a review. However, the coverage of such prediction intervals relies heavily on the parametric model being correct (well-specified), and can even fail in certain high-dimensional regimes where the model is indeed correct (Bai et al., 2021).

## 2 PRELIMINARIES

**Uncertainty quantification via prediction sets**    We consider standard learning problems in which we observe a dataset $D$ of examples $(x_i, y_i) \in \mathcal{X} \times \mathcal{Y}$ from some data distribution, and wish to predict the label $y$ from the input $x$. A *prediction set* is a set-valued function $C : \mathcal{X} \to 2^{\mathcal{Y}}$ where $C(x)$ is a subset of $\mathcal{Y}$. Two prevalent examples are regression ($\mathcal{Y} = \mathbb{R}$) in which we can choose $C(x) \subset \mathbb{R}$ as a **prediction interval**, and (multi-class) classification ($\mathcal{Y} = [L] := \{1, \dots, L\}$) in which we can choose $C(x) \subset [L]$ as a (discrete) **label prediction set**.

**Coverage and efficiency** The (marginal) coverage probability (henceforth *coverage*) of a prediction set $C$ is defined as

$$\text{Coverage}(C) := \mathbb{P}(Y \in C(X))$$

where $(X, Y)$ is a test example from the same data distribution. We also define the (mis)-coverage loss $L_{\text{coverage}}(C) := 1 - \text{Coverage}(C) = \mathbb{P}(Y \notin C(X))$. A learned prediction set is often desired to achieve *valid coverage* in the sense that $\text{Coverage}(C) \geq 1 - \alpha$ for some $\alpha \in (0, 1)$. Here $1 - \alpha$ is a pre-determined target coverage level; typical choices are e.g. $1 - \alpha \in \{90\%, 95\%\}$, which corresponds to picking $\alpha \in \{0.1, 0.05\}$.

In addition to valid coverage, it is often desired that the prediction set has a good *efficiency* (such as small size). This is motivated by the fact that valid coverage can be achieved trivially if we do not care about the size, e.g. by always outputting $C = \mathcal{Y}$, which is not informative. Throughout this paper we will use $\ell_{\text{eff}}$ to denote the particular efficiency loss we care about, where $\ell_{\text{eff}}(C; (x, y))$ measures the efficiency loss of $C$ on an example $(x, y)$, such as the length (Lebesgue measure) of prediction intervals, or the size (cardinality) of label prediction sets.

**Nested set framework** We adopt the nested set framework of (Gupta et al., 2019) for convenience for our presentation and analysis. A family $\{C_t\}_{t \in \mathcal{T} \subset \mathbb{R}}$ is said to be a (family of) nested sets if $t \leq t'$ implies that $C_t(x) \subset C_{t'}(x)$ for all $x \in \mathcal{X}$. Throughout this paper out notation $C_t$ or $C_{\theta,t}$ are assumed to be nested sets with respect to $t$. We assume that our efficiency loss $\ell_{\text{eff}}$ is non-decreasing w.r.t. its (set-valued) argument, i.e. $\ell_{\text{eff}}(C; (x, y)) \leq \ell_{\text{eff}}(C'; (x, y))$ if $C \subseteq C'$. Therefore, for nested sets the loss $t \mapsto \ell_{\text{eff}}(C_t; (x, y))$ is non-decreasing in $t$. As the coverage loss $L(C_t) = \mathbb{P}(Y \notin C_t(X))$ (and its empirical version) is instead non-increasing in $t$, the efficiency loss and the coverage loss always comes as a trade-off.

## 2.1 CONFORMAL PREDICTION

Conformal prediction (Vovk et al., 2005; Lei & Wasserman, 2014) is a powerful technique for learning prediction sets with coverage guarantees. The core of conformal prediction is its *conformalization* step, which turns any base prediction function (or training algorithm) into a prediction set.

We here briefly review conformal prediction using the vanilla (split) conformal regression method of (Lei et al., 2018), and refer the readers to (Angelopoulos & Bates, 2021) for more examples. Given any *base predictor* $f : \mathcal{X} \to \mathbb{R}$ (potentially learned on a training dataset $D_{\text{train}}$), conformal prediction outputs a prediction interval

$$C_{\widehat{t}}(x) := \left[ f(x) - \widehat{t}, f(x) + \widehat{t} \right], \tag{1}$$

where $\widehat{t} \in \mathbb{R}_{\geq 0}$ is chosen as the $(1 - \alpha)$-quantile[1] of $|y - f(x)|$ on a calibration dataset $D_{\text{cal}}$ with size $n_{\text{cal}} := |D_{\text{cal}}|$ using the following conformalization step:

$$\widehat{t} = \lceil (1 - \alpha) n_{\text{cal}} \rceil \text{-th largest of } \{|y_i - f(x_i)|\}_{i=1}^{n_{\text{cal}}}. \tag{2}$$

The main guarantee for the learned interval $C_{\widehat{t}}$ is that it achieves a $(1 - \alpha)$ coverage guarantee of the form $\mathbb{P}_{D_{\text{cal}},(X,Y)}(Y \in C_{\widehat{t}}(X)) \geq 1 - \alpha$ (Lei et al., 2018, Theorem 2.2). The proof relies on the exchangeability between the scores $\{|y_i - f(x_i)|\}_{i=1}^{n_{\text{cal}}}$ and $|Y - f(X)|$, which allows this guarantee to hold in a distribution-free fashion (i.e. for any data distribution).

**Conformal prediction as a constrained ERM with one parameter** We start by a simple reinterpretation that the conformalization step (2) is equivalent to solving a constrained empirical risk minimization (ERM) problem with a single learnable parameter $t$ (cf. Appendix A for the proof).

**Proposition 1** (Conformal regression as a constrained ERM with one learnable parameter)**.** *The parameter* $\widehat{t} \in \mathbb{R}$ *defined in* (2) *is the solution to the following constrained ERM problem*

$$
\begin{aligned}
&\underset{t \geq 0}{\text{minimize}} \ \widehat{L}_{\text{eff}}(C_t) := \frac{1}{n_{\text{cal}}} \sum_{i \in D_{\text{cal}}} \ell_{\text{eff}}(C_t; (x_i, y_i)) = 2t \\
&\text{subject to} \ \widehat{L}_{\text{coverage}}(C_t) := \frac{1}{n_{\text{cal}}} \sum_{i \in D_{\text{cal}}} \mathbf{1}\{y_i \notin C_t(x_i)\} \leq \alpha.
\end{aligned}
\tag{3}
$$

---

[1]Technically (2) requires the $\lceil (1 - \alpha)(n_{\text{recal}} + 1) \rceil$-th largest element to guarantee valid coverage (Vovk et al., 2005); here we choose the close $\lceil (1 - \alpha) n_{\text{recal}} \rceil$-th largest to allow the following insight.

---

**Algorithm 1** Conformal Prediction with General Function Class (CP-Gen)

---

**Input:** Class of prediction sets $\mathcal{C} = \{C_{\theta,t}\}_{\theta \in \Theta, t \in \mathcal{T}}$; target miscoverage level $\alpha \in (0,1)$; $\varepsilon_0 \geq 0$.
   Efficiency loss $\ell_{\mathrm{eff}}$; Calibration dataset $D_{\mathrm{cal}}$ with size $n_{\mathrm{cal}}$.
1: Solve the following constrained ERM problem on dataset $D_{\mathrm{cal}}$ (with relaxation parameter $\varepsilon_0$):

$$
\begin{aligned}
(\widehat{\theta}, \widehat{t}) &\leftarrow \underset{\theta \in \Theta, t \in \mathcal{T}}{\arg\min} \ \widehat{L}_{\mathrm{eff}}(C_{\theta,t}) := \frac{1}{n_{\mathrm{cal}}} \sum_{i \in D_{\mathrm{cal}}} \ell_{\mathrm{eff}}(C_{\theta,t}(x_i), y_i) \\
&\text{subject to } \ \widehat{L}_{\mathrm{coverage}}(C_{\theta,t}) := \frac{1}{n_{\mathrm{cal}}} \sum_{i \in D_{\mathrm{cal}}} \mathbf{1}\{y_i \notin C_{\theta,t}(x_i)\} \leq \alpha + \varepsilon_0.
\end{aligned}
\tag{5}
$$

**Output:** Prediction set $C_{\widehat{\theta}, \widehat{t}}$.

---

*Above, $\ell_{\mathrm{eff}}(C; (x,y)) = \mathrm{length}(C(x))$ is the length of the interval $C(x)$.*

Though simple, this re-interpretation suggests a limitation to the conformalization step (2) as well as its analogue in other existing conformal methods: It only learns a single parameter $t$, and thus cannot further optimize the efficiency due to the coverage-efficiency trade-off (cf. Figure 1). However, the form of the constrained ERM problem (3) suggests that it can be readily extended to more general function classes with more than one learnable parameters, which is the focus of this work.

## 3 CONFORMAL PREDICTION WITH GENERAL FUNCTION CLASSES

### 3.1 ALGORITHM

Our algorithm, Conformal Prediction with General Function Classes (CP-Gen; full description in Algorithm 1), is an extension of the constrained ERM problem (3) into the case of general function classes with multiple learnable parameters. CP-Gen takes in a function class of prediction sets

$$
\mathcal{C} := \{C_{\theta,t}(x) : \theta \in \Theta, t \in \mathcal{T} \subset \mathbb{R}\},
\tag{4}
$$

where (as mentioned) we assume that $\{C_{\theta,t}\}_{t \in \mathcal{T}}$ is a nested set for each $\theta \in \Theta$. The parameter set $\Theta$ as well as the form of $C_{\theta,t}$ in (4) can be arbitrary, depending on applications and the available base predictors at hand. Given $\mathcal{C}$, our algorithm then solves the constrained ERM problem (5) of finding the smallest interval among $\mathcal{C}$ subject to valid coverage on dataset $D_{\mathrm{cal}}$.

Compared with vanilla conformal prediction, Algorithm 1 allows more general tasks with an arbitrary function class and efficiency loss; for example, this encompasses several recent algorithms such as finite hyperparameter selection and linear aggregation (Yang & Kuchibhotla, 2021; Chen et al., 2021). We remark that (5) includes an additional relaxation parameter $\varepsilon_0 \geq 0$ for the coverage constraint. This is for analysis (for Proposition 2(b) & 7(b)) only; our implementation uses $\varepsilon_0 = 0$.

### 3.2 THEORY

An important theoretical question about CP-Gen is whether it achieves coverage and efficiency guarantees on the population (test data). This section showcases that, by standard generalization arguments, CP-Gen achieves approximate validity and near-optimal efficiency whenever function class is low-capacity in a certain sense. We remark that our experiments use the modified algorithm CP-Gen-Recal (Section 3.3) which involves a reconformalization step. Here we focus on CP-Gen as we believe its theory could be more informative.

Let $L_{\{\mathrm{eff}, \mathrm{coverage}\}}(C_{\theta,t}) := \mathbb{E}[\ell_{\{\mathrm{eff}, \mathrm{coverage}\}}(C_{\theta,t}; (X, Y))]$ denote the population coverage and efficiency losses for any $(\theta, t)$. We define the following uniform concentration quantities:

$$
\varepsilon_{\mathrm{eff}} := \sup_{\theta \in \Theta, t \in \mathcal{T}} \left| \widehat{L}_{\mathrm{eff}}(C_{\theta,t}) - L_{\mathrm{eff}}(C_{\theta,t}) \right|,
\tag{6}
$$

$$
\varepsilon_{\mathrm{coverage}} := \sup_{\theta \in \Theta, t \in \mathcal{T}} \left| \widehat{L}_{\mathrm{coverage}}(C_{\theta,t}) - L_{\mathrm{coverage}}(C_{\theta,t}) \right|.
\tag{7}
$$

---

**Algorithm 2** Conformal Prediction with General Fn. Class and Recalibration (`CP-Gen-Recal`)

---

**Input:** Class of prediction sets $\mathcal{C} = \{C_{\theta,t}\}_{\theta \in \Theta, t \in \mathcal{T}}$; target miscoverage level $\alpha \in (0,1)$; $\varepsilon_0 \geq 0$.
Efficiency loss $\ell_{\mathrm{eff}}$; Calibration datasets $D_{\mathrm{cal}}, D_{\mathrm{recal}}$ with size $n_{\mathrm{cal}}, n_{\mathrm{recal}}$.

1: Run Algorithm 1 on dataset $D_{\mathrm{cal}}$ (with relaxation parameter $\varepsilon_0$) to obtain $(\widehat{\theta}, \widehat{t})$.
2: Keep $\widehat{\theta}$, and reconformalize $t \in \mathcal{T}$ on the recalibration dataset $D_{\mathrm{recal}}$:

$$\widehat{t}_{\mathrm{recal}} \leftarrow \inf\left\{ t \in \mathcal{T} : y_i \in C_{\widehat{\theta},t}(x_i) \text{ for at least } \lceil (1-\alpha)(n_{\mathrm{recal}} + 1) \rceil \text{ examples } (x_i, y_i) \in D_{\mathrm{recal}} \right\}.$$

**Output:** Prediction set $C_{\widehat{\theta}, \widehat{t}_{\mathrm{recal}}}$.

---

The following proposition connects the generalization of `CP-Gen` to the above uniform concentration quantities by standard arguments (see Appendix B for the proof. We remark that the proof relies on $D_{\mathrm{cal}}$ being i.i.d., which is slightly stronger than exchangeability assumption commonly assumed in the conformal prediction literature.)

**Proposition 2** (Generalization of `CP-Gen`). *The prediction set $C_{\widehat{\theta},\widehat{t}}$ learned by Algorithm 1 satisfies*

*(a) (Approximately valid population coverage) We have*

$$L_{\mathrm{coverage}}(C_{\widehat{\theta},\widehat{t}}) \leq \alpha + \varepsilon_0 + \varepsilon_{\mathrm{coverage}},$$

*i.e. the population coverage of $C_{\widehat{\theta},\widehat{t}}$ is at least $1 - \alpha - (\varepsilon_0 + \varepsilon_{\mathrm{coverage}})$.*

*(b) (Near-optimal efficiency) Suppose $\varepsilon_0 \geq \varepsilon_{\mathrm{coverage}}$, then we further have*

$$L_{\mathrm{eff}}(C_{\widehat{\theta},\widehat{t}}) \leq \inf_{\substack{(\theta,t) \in \Theta \times \mathcal{T} \\ L_{\mathrm{coverage}}(C_{\theta,t}) \leq \alpha}} L_{\mathrm{eff}}(C_{\theta,t}) + 2\varepsilon_{\mathrm{eff}},$$

*i.e. $C_{\widehat{\theta},\widehat{t}}$ achieves $2\varepsilon_{\mathrm{eff}}$-near-optimal efficiency against any prediction set within $\mathcal{C}$ with at least $(1-\alpha)$ population coverage.*

**Examples of good generalization** Proposition 2 shows that `CP-Gen` acheives approximate coverage and near-optimal efficiency if the concentration terms $\varepsilon_{\mathrm{eff}}$ and $\mathrm{eff}_{\mathrm{coverage}}$ are small. In Appendix C, we bound these on two example function classes: *Finite Class* (Proposition 4) and *VC/Rademacher Class* (Proposition 5). Both classes admit bounds of the form $\{\varepsilon_{\mathrm{eff}}, \mathrm{eff}_{\mathrm{coverage}}\} \leq \sqrt{\mathrm{Comp}(\mathcal{C})/n_{\mathrm{cal}}}$ with high probability via standard concentration arguments, where $\mathrm{Comp}(\mathcal{C})$ is a certain complexity measure of $\mathcal{C}$. Combined with Proposition 2, our `CP-Gen` algorithm with these classes achieve an $1 - \alpha - \sqrt{\mathrm{Comp}(\mathcal{C})/n_{\mathrm{cal}}}$ approximate coverage guarantee and $\sqrt{\mathrm{Comp}(\mathcal{C})/n_{\mathrm{cal}}}$ near-optimal efficiency guarantee. In particular, our Proposition 4 recovers the coverage guarantee for the finite-class selection algorithm of (Yang & Kuchibhotla, 2021, Theorem 1) though our efficiency guarantee is more general.

We remark that both examples above contain important applications. The finite class contains e.g. optimizing over a $K$-dimensional hyperparameter to use via grid search, with e.g. $(1/\delta)^K$ confidence sets and thus $\mathrm{Comp}(\mathcal{C}) = O(\log((1/\delta)^K)) = O(K \log(1/\delta))$. The VC/Rademacher class contains the important special case of linear classes with $K$ base predictors (we defer the formal statement and proof to Appendix C.3). Also, these examples are not necessarily exhaustive. Our take-away is rather that we may expect `CP-Gen` to generalize well (and thus achieves good coverage and efficiency) more broadly in practice, for instance whenever it learns $K \ll n_{\mathrm{cal}}$ parameters.

## 3.3 ALGORITHM WITH VALID COVERAGE VIA RECONFORMALIZATION

Although `CP-Gen` enjoys theoretical bounds on the coverage and efficiency, a notable drawback is that it does not guarantee *exactly valid* (at least) $1 - \alpha$ coverage like usual conformal prediction, and its approximate coverage bound depends on the uniform concentration quantity $\varepsilon_{\mathrm{coverage}}$ that is not computable from the observed data without structural assumptions on the function class $\mathcal{C}$.

To remedy this, we incorporate a simple reconformalization technique on another recalibration dataset $D_{\mathrm{recal}}$ (e.g. a further data split), which guarantees valid finite-sample coverage by exchangeability. We call this algorithm `CP-Gen-Recal` and provide the full description in Algorithm 2.

We remark that this reconformalization technique for obtaining guaranteed $1 - \alpha$ coverage is widely used in the conformal prediction literature, e.g. (Angelopoulos et al., 2020). However, to the best of our knowledge, there is no known analysis for our `CP-Gen-Recal` algorithm for general function classes, for which we provide a result below (formal statement and proof can be found in Proposition 7 & Appendix D). The proof of the coverage bound is standard as in the conformal prediction literature, while the proof of the efficiency bound builds upon the result for `CP-Gen` (Proposition 2(b)) and handles additional concentration terms from the reconformalization step.

**Proposition 3** (Coverage and efficiency guarantee for `CP-Gen-Recal`; Informal version). *The prediction set $C_{\widehat{\theta}, \widehat{t}_{\mathrm{recal}}}$ learned by Algorithm 2 achieves $(1 - \alpha)$ finite-sample coverage: $\mathbb{P}_{D_{\mathrm{recal}}, (X, Y)}(Y \in C_{\widehat{\theta}, \widehat{t}_{\mathrm{recal}}}) \geq 1 - \alpha$. Further, it achieves $O(\varepsilon_{\mathrm{eff}} + \varepsilon_{\mathrm{coverage}} + 1/\sqrt{n_{\mathrm{recal}}})$ near-optimal efficiency under additional regularity assumptions.*

## 4 DIFFERENTIABLE OPTIMIZATION

Our (meta) algorithms `CP-Gen` and `CP-Gen-Recal` involve solving the constrained ERM problem (5). One feasible case is when $\Theta$ is finite and small, in which we enumerate all possible $\theta \in \Theta$ and find the optimal $\widehat{t}$ for each $\theta$ efficiently using quantile computation. However, this optimization is significantly more challenging when the underlying parameter set $\Theta$ is continuous and we wish to jointly optimize over $(\theta, t)$: The coverage loss $\widehat{L}_{\mathrm{coverage}}(C_{\theta, t})$ is *non-differentiable* and its "gradient" is zero almost everywhere as it uses the zero-one loss. This makes the coverage constraint challenging to deal with and not amenable to any gradient-based algorithm.

To address this non-differentiability, we develop a gradient-based practical implementation by approximating the coverage constraint. We first rewrite each individual coverage loss into the form
$$\mathbf{1}\{y \notin C_{\theta, t}(x)\} = \mathbf{1}\{s_{\theta, t}(x, y) < 0\} = \ell_{01}(s_{\theta, t}(x, y)).$$
where $\ell_{01}(z) := \mathbf{1}\{z < 0\}$ is the zero-one loss. (Such a rewriting is possible in most cases by taking $s_{\theta, t}$ as a suitable "score-like" function; see Appendix E for instantiations in our experiments.) Then, inspired by the theory of surrogate losses (Bartlett et al., 2006), we approximate $\ell_{01}(z)$ by the hinge loss $\ell_{\mathrm{hinge}}(z) = [1 - z]_{+}$ which is (almost everywhere) differentiable with a non-trivial gradient. We find the hinge loss to perform better empirically than alternatives such as the logistic loss.

To deal with the (modified) constraint, we turn it into an exact penalty term with penalty parameter $\lambda \geq 0$, and use a standard primal-dual formulation (Bertsekas, 1997) to obtain an unconstrained min-max optimization problem on the Lagrangian:
$$\min_{\theta, t} \max_{\lambda \geq 0} \widehat{L}_{\mathrm{eff}}(C_{\theta, t}) + \lambda \left[ \widehat{L}_{\mathrm{hinge}}(C_{\theta, t}) - \alpha \right]_{+}, \tag{8}$$
where $\widehat{L}_{\mathrm{hinge}}(C_{\theta, t}) := \frac{1}{n_{\mathrm{cal}}} \sum_{i=1}^{n_{\mathrm{cal}}} \ell_{\mathrm{hinge}}(s_{\theta, t}(x_i, y_i))$ is the empirical hinge loss on the calibration dataset $D_{\mathrm{cal}}$. Our final practical implementation of (5) solves the problem (8) by Stochastic Gradient Descent-Ascent (with respect to $(\theta, t)$ and $\lambda$) to yield an approximate solution $(\widehat{\theta}, \widehat{t})$. We remark that in our experiments where we use the reconformalized version `CP-Gen-Recal`, we only keep the $\widehat{\theta}$ obtained from (8) and perform additional reconformalization to compute $\widehat{t}_{\mathrm{recal}}$ to guarantee coverage.

We also emphasize that the approximation in (8) makes the problem differentiable at the cost of deviating from the true constrained ERM problem (5) and thus potentially may sacrifice in terms of the efficiency. However, our experiments in Section 5 show that such an implementation can still improve the efficiency over existing approaches in practice, despite the approximation.

## 5 EXPERIMENTS

We empirically test our `CP-Gen-Recal` algorithm (using the practical implementation (8)) on three representative real-data tasks. The concrete construction of $\{C_{\theta, t}\}$ will be described within each application. Throughout this section we choose $1 - \alpha = 90\%$ as the nominal coverage level, and use the `CP-Gen-Recal` algorithm to guarantee coverage in expectation. We provide ablations with $\alpha \in \{80\%, 95\%\}$ in Appendix G.1 and G.2 and the `CP-Gen` algorithm in Appendix G.3. Several additional ablations and analyses can be found in Appendix H.

Table 1: **Results for conformal quantile finetuning** on real-data regression tasks at level $1 - \alpha = 90\%$. For each method we report the (test) coverage, length, and pinball loss of the corresponding base quantile predictor. All results are averaged over 8 random seeds.

| Dataset | CQR | | | QR + CP-Gen-Recal (ours) | | |
|---|---|---|---|---|---|---|
| | Coverage(%) | Length | $L_{\text{pinball}}^{\text{test}}$ | Coverage(%) | Length | $L_{\text{pinball}}^{\text{test}}$ |
| MEPS_19 | 89.98 | 1.167 | 0.112 | 90.09 | **0.890** | 0.131 |
| MEPS_20 | 89.72 | 1.165 | 0.117 | 89.99 | **0.830** | 0.141 |
| MEPS_21 | 89.81 | 1.145 | 0.107 | 90.22 | **0.962** | 0.129 |
| Facebook_1 | 90.12 | 0.555 | 0.052 | 90.34 | **0.384** | 0.090 |
| Facebook_2 | 90.13 | 0.491 | 0.044 | 90.02 | **0.364** | 0.092 |
| kin8nm | 90.03 | 1.214 | 0.076 | 89.31 | **1.173** | 0.078 |
| naval | 89.70 | 3.095 | 0.164 | 89.71 | **3.077** | 0.166 |
| bio | 90.26 | 2.271 | 0.130 | 90.20 | **2.164** | 0.148 |
| blog_data | 90.19 | 0.605 | 0.058 | 90.01 | **0.496** | 0.107 |
| Nominal $(1 - \alpha)$ | 90.00 | - | - | 90.00 | - | - |

## 5.1 IMPROVED PREDICTION INTERVALS VIA CONFORMAL QUANTILE FINETUNING

**Setup**  We consider regression tasks in which we use quantile regression (pinball loss) to train a base quantile predictor $\widehat{F}(x) = [\widehat{f}_{\text{lo}}(x), \widehat{f}_{\text{hi}}(x)] = \widehat{\theta}_0^\top \widehat{\Phi}(x)$ on $D_{\text{train}}$ (with learning rate decay by monitoring validation loss on $D_{\text{cal}}$). Here $\widehat{\Phi} : \mathcal{X} \to \mathbb{R}^{d_h}$ is the learned representation function, $\widehat{\theta}_0 \in \mathbb{R}^{d_h \times 2}$ is the last linear layer ($d_h$ denotes the last hidden dimension), and $\widehat{f}_{\text{lo}}, \widehat{f}_{\text{hi}}$ are the learned {lower, upper} quantile functions (see Appendix E.1 for more details on the training procedure). Given $\widehat{F}$, we learn a baseline prediction interval of the form $[\widehat{f}_{\text{lo}}(x) - t, \widehat{f}_{\text{hi}}(x) + t]$ on $D_{\text{recal}}$ via Conformalized Quantile Regression (CQR) (Romano et al., 2019).

We then attempt to improve the length over CQR by *conformal quantile finetuning*: Fix the representation function $\widehat{\Phi}$ and finetune the linear layer $\theta$ using our CP-Gen-Recal algorithm, so that $C_{\theta,t}(x) = [\theta_{\text{lo}}^\top \widehat{\Phi}(x) - t, \theta_{\text{hi}}^\top \widehat{\Phi}(x) + t]$ (where $\theta = [\theta_{\text{lo}}, \theta_{\text{hi}}]$). We learn a new $\widehat{\theta}$ on $D_{\text{cal}}$ via (8) (where $\ell_{\text{eff}}$ is chosen as the length), and then compute $\widehat{t}_{\text{recal}}$ on $D_{\text{recal}}$ as in Algorithm 2.

We perform the above on 9 real-world regression datasets with a 3-layer MLP with width $d_h = 64$, similar as (Romano et al., 2019; Feldman et al., 2021). Additional details about the setup can be found in Appendix E.1. We also test various tweaks of the CQR baseline (results provided in Appendix H.2).

**Results**  Table 1 compares the (test) coverage and length between CQR and the finetuned linear layer via our CP-Gen-Recal. While both CQR and CP-Gen-Recal achieves valid 90% coverage, CP-Gen-Recal can systematically improve the length over CQR on all tasks. Table 1 also reports the pinball loss for both the base $\widehat{\theta}_0 \widehat{\Phi}(x)$ as well as the fine-tuned $\widetilde{\theta}^\top \widehat{\Phi}(x)$ on the test set $D_{\text{test}}$. Intriguingly, our conformal finetuning made the pinball loss *worse* while managing to improve the length. This suggests the unique advantage of our constrained ERM objective, as it rules out the simple explanation that the length improvement is just because of a lower test loss. We remark that while CP-Gen-Recal improves the length over CQR, it comes at a cost in terms of worse conditional coverage (analysis presented in Appendix H.1).

## 5.2 MINIMUM-VOLUME PREDICTION SETS FOR MULTI-OUTPUT REGRESSION

**Setup**  This task aims to learn a box-shaped prediction set for multi-output regression with a *small volume*. Our learning task is regression with output dimension $d_{\text{out}} > 1$. We first learn a based predictor $\widehat{f} : \mathbb{R}^d \to \mathbb{R}^{d_{\text{out}}}$ by minimizing the MSE loss on $D_{\text{train}}$. We then learn a *box-shaped* prediction set of the form $C_u(x) = \prod_{i=1}^{d_{\text{out}}} [\widehat{f}_i(x) - \widehat{u}_i, \widehat{f}_i(x) + \widehat{u}_i]$ by one of the following methods:

- (Coord-wise): Each $\widehat{u}_i$ is obtained by vanilla conformalization (2) over the $i$-th output coordinate on $D_{\text{cal}} \cup D_{\text{recal}}$. To guarantee $1 - \alpha$ coverage, each coordinate is conformalized at level $1 - \alpha/d_{\text{out}}$, motivated by the union bound.

Table 2: **Results for multi-output regression** on next-state prediction tasks, at level $1 - \alpha = 90\%$. For each method we report the (test) coverage and volume of its learned box-shaped prediction set. The reported volume is the "halfened" version $\prod_{i=1}^{d_{\text{out}}} u_i$. All results are averaged over 8 random seeds.

| Dataset | Coord-wise | | Coord-wise-Recal | | CP-Gen-Recal (ours) | |
|---|---|---|---|---|---|---|
| | Coverage(%) | Volume | Coverage(%) | Volume | Coverage(%) | Volume |
| Cartpole | 94.28 | $1.20 \times 10^{-5}$ | 90.17 | $5.10 \times 10^{-6}$ | 90.12 | $\mathbf{2.30 \times 10^{-6}}$ |
| Half-Cheetah | 93.90 | $1.10 \times 10^{-5}$ | 90.06 | $1.23 \times 10^{-6}$ | 90.02 | $\mathbf{9.07 \times 10^{-7}}$ |
| Ant | 93.56 | $3.37 \times 10^{-3}$ | 89.99 | $1.70 \times 10^{-4}$ | 90.02 | $\mathbf{8.25 \times 10^{-5}}$ |
| Walker | 94.42 | $2.59 \times 10^{-5}$ | 90.01 | $7.33 \times 10^{-7}$ | 89.94 | $\mathbf{3.47 \times 10^{-7}}$ |
| Swimmer | 95.62 | $2.80 \times 10^{-5}$ | 89.90 | $2.22 \times 10^{-6}$ | 90.13 | $\mathbf{1.46 \times 10^{-7}}$ |
| Hopper | 92.87 | $2.81 \times 10^{-9}$ | 90.02 | $1.01 \times 10^{-9}$ | 89.92 | $\mathbf{8.25 \times 10^{-10}}$ |
| Humanoid | 94.75 | $4.28 \times 10^{-4}$ | 89.95 | $8.53 \times 10^{-8}$ | 89.94 | $\mathbf{4.95 \times 10^{-8}}$ |
| Nominal $(1 - \alpha)$ | 90.00 | - | 90.00 | - | 90.00 | - |

- (Coord-wise-Recal): Perform the above on $D_{\text{cal}}$ to learn $\widehat{u}_i$, and reconformalize an additional $t \geq 0$ on $D_{\text{recal}}$ to reshape the prediction set *proportionally*:

$$C_{\widehat{u},t}(x) = \prod_{i=1}^{d_{\text{out}}} [\widehat{f}_i(x) - t\widehat{u}_i, \widehat{f}_i(x) + t\widehat{u}_i]. \tag{9}$$

- (CP-Gen-Recal, ours): Optimize the volume directly over all $u \in \mathbb{R}^{d_{\text{out}}}$ using (8) on $D_{\text{cal}}$, where $\ell_{\text{eff}}(C_u; (x, y)) = \prod_{i=1}^{d_{\text{out}}} (2u_i)$ is chosen as the volume loss. We then reconformalize an additional $\widehat{t} \geq 0$ on $D_{\text{recal}}$ to reshape the prediction set same as in (9). Note that this reconformalization step is equivalent to Algorithm 2 with the re-parametrization $\widehat{u} \mapsto (\widehat{\theta}, \widehat{t})$ where $\widehat{\theta} \in \mathbb{R}_{>0}^{d_{\text{out}}-1}$ denotes the ratio between $\widehat{u}_i$, and $\widehat{t} \in \mathbb{R}_{>0}$ denotes a common scale.

Our datasets are a collection of *next-state prediction tasks* with multi-dimensional continuous states in offline reinforcement learning (RL), constructed similarly as D4RL (Fu et al., 2020) with some differences. Additional details about the dataset and experimental setup are in Appendix E.2. We also test an additional Max-score-Conformal baseline (which uses vanilla conformal prediction with score function $\|y - \widehat{f}(x)\|_\infty$, equivalent to a *hypercube*-shaped predictor) in Appendix H.3, which we find also performs worse than our CP-Gen-Recal.

**Results** Table 2 reports the (test) coverage and volume of the above three methods. The Coord-wise method achieves valid coverage but is quite conservative (over-covers), which is as expected as the union bound is worst-case in nature and the coordinate-wise conformalization does not utilize the potential correlation between the output coordinates. Coord-wise-Recal achieves approximately 90% coverage with a much smaller volume. Our CP-Gen-Recal also achieves valid 90% coverage but a further lower volume across all tasks. This suggests that optimizing the volume over all possible $u \in \mathbb{R}^{d_{\text{out}}}$ data-dependently using our CP-Gen-Recal is indeed more flexible than pre-determined conformalization schemes such as Coord-wise.

**Additional experiment: label prediction sets for ImageNet** We show that CP-Gen-Recal can learn label prediction sets for ImageNet with valid coverage and improved size over existing approaches, by finding an optimized set of ensemble weights over multiple base neural networks (Table 5). The full setup and results are presented in Appendix F.

## 6 CONCLUSION

This paper proposes Conformal Prediction with General Function Class, a conformal prediction algorithm that optimizes the efficiency metric subject to valid coverage over a general function class of prediction sets. We provide theoretical guarantees for its coverage and efficiency in certain situations, and develop a gradient-based practical implementation which performs well empirically on several large-scale tasks. We believe our work opens up many directions for future work, such as stronger theoretical guarantees via more structured function classes, further improving the gradient-based approximate implementation, or experiments on other uncertainty quantification tasks.

## ACKNOWLEDGMENT

We thank Silvio Savarese for the suggestion on the multi-output regression task, and Yuping Luo for the help with preparing the offline reinforcement learning dataset. We thank the anonymous reviewers for their valuable feedback.

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

## A    PROOF OF PROPOSITION 1

Recall that $C_t(x) = [f(x) - t, f(x) + t]$ satisfies $\ell_{\text{eff}}(C_t; (x, y)) = \text{length}(C_t(x)) = 2t$ for any $(x, y)$. Also, we have

$$\mathbf{1}\left\{y \notin C_t(x)\right\} = \mathbf{1}\left\{y \notin [f(x) - t, f(x) + t]\right\} = \mathbf{1}\left\{|y - f(x)| > t\right\},$$

or equivalently $\mathbf{1}\left\{y \in C_t(x)\right\} = \mathbf{1}\left\{|y - f(x)| \leq t\right\}$. Therefore, problem (3) is equivalent to

$$\operatorname*{minimize}_{t \geq 0}\ t$$

$$\text{subject to}\ \ 1 - \widehat{L}_{\text{coverage}}(C_t) := \frac{1}{n_{\text{cal}}} \sum_{i=1}^{n_{\text{cal}}} \mathbf{1}\left\{|y_i - f(x_i)| \leq t\right\} \geq 1 - \alpha.$$

By definition of quantiles, this problem is solved at

$$\widehat{t} = \lceil (1 - \alpha) n_{\text{cal}} \rceil \text{ -th largest element of } \{|y_i - f(x_i)|\}_{i=1}^{n_{\text{cal}}},$$

which is the desired result. $\qquad\square$

## B    PROOF OF PROPOSITION 2

(a) As $C_{\widehat{\theta},\widehat{t}}$ solves problem (5), it satisfies the constraint $\widehat{L}_{\text{coverage}}(C_{\widehat{\theta},\widehat{t}}) \leq \alpha + \varepsilon_0$. Therefore,

$$L_{\text{coverage}}(C_{\widehat{\theta},\widehat{t}}) = \underbrace{\widehat{L}_{\text{coverage}}(C_{\widehat{\theta},\widehat{t}})}_{\leq \alpha + \varepsilon_0} + L_{\text{coverage}}(C_{\widehat{\theta},\widehat{t}}) - \widehat{L}_{\text{coverage}}(C_{\widehat{\theta},\widehat{t}})$$

$$\leq \alpha + \varepsilon_0 + \sup_{(\theta,t) \in \Theta \times \mathcal{T}} \left| L_{\text{coverage}}(C_{\theta,t}) - \widehat{L}_{\text{coverage}}(C_{\theta,t}) \right|$$

$$= \alpha + \varepsilon_0 + \varepsilon_{\text{coverage}}.$$

(b) Suppose $\varepsilon_{\text{coverage}} \leq \varepsilon_0$. Taking any $(\theta, t) \in \Theta \times \mathcal{T}$ such that $L_{\text{coverage}}(C_{\theta,t}) \leq \alpha$, we have

$$\widehat{L}_{\text{coverage}}(C_{\theta,t}) \leq L_{\text{coverage}}(C_{\theta,t}) + \varepsilon_{\text{coverage}} \leq \alpha + \varepsilon_{\text{coverage}} \leq \alpha + \varepsilon_0.$$

This shows that $(\theta, t)$ lies within the constraint set of problem (5). Thus as $(\widehat{\theta}, \widehat{t})$ further minimizes the loss $\widehat{L}_{\text{eff}}$ within the constraint set, we have $\widehat{L}_{\text{eff}}(C_{\widehat{\theta},\widehat{t}}) \leq \widehat{L}_{\text{eff}}(C_{\theta,t})$. This shows that

$$L_{\text{eff}}(C_{\widehat{\theta},\widehat{t}}) - L_{\text{eff}}(C_{\theta,t})$$

$$\leq \underbrace{\widehat{L}_{\text{eff}}(C_{\widehat{\theta},\widehat{t}}) - \widehat{L}_{\text{eff}}(C_{\theta,t})}_{\leq 0} + 2 \sup_{(\theta,t)\in\Theta\times\mathcal{T}} \left|\widehat{L}_{\text{eff}}(C_{\theta,t}) - L_{\text{eff}}(C_{\theta,t})\right|$$

$$\leq 2\varepsilon_{\text{eff}}.$$

As the above holds simultaneously for all $(\theta, t) \in \Theta \times \mathcal{T}$ with at most $\alpha$ coverage loss, taking the sup over the left-hand side yields

$$L_{\text{eff}}(C_{\widehat{\theta},\widehat{t}}) - \inf_{\substack{(\theta,t)\in\Theta\times\mathcal{T} \\ L_{\text{coverage}}(C_{\theta,t})\leq\alpha}} L_{\text{eff}}(C_{\theta,t}) \leq 2\varepsilon_{\text{eff}}.$$

$\square$

## C   EXAMPLES OF GOOD GENERALIZATION FOR CP-Gen

We provide two concrete examples where the concentration terms $\varepsilon_{\text{eff}}$ and $\varepsilon_{\text{coverage}}$ are small with high probability, in which case Proposition 2 guarantees that CP-Gen learns an prediction set with approximate validity and near-optimal efficiency.

**Assumption A** (Bounded and Lipschitz efficiency loss). *The loss function $\ell_{\text{eff}}$ satisfies*

*A1 (Bounded loss). $|\ell_{\text{eff}}(C_{\theta,t}; (x,y))| \leq M$ for all $(\theta,t) \in \Theta \times \mathcal{T}$ and all $(x,y) \in \mathcal{X} \times \mathcal{Y}$.*

*A2 (t-Lipschitzness). $t \mapsto \ell_{\text{eff}}(C_{\theta,t}; (x,y))$ is $L_{\mathcal{T}}$-Lipschitz for all $(\theta,x,y) \in \Theta \times \mathcal{X} \times \mathcal{Y}$.*
**Assumption B** (Bounded $\mathcal{T}$). *The parameter space $\mathcal{T} \subset \mathbb{R}$ is bounded: $\sup_{t\in\mathcal{T}} |t| \leq B_{\mathcal{T}}$.*

### C.1   FINITE CLASS

**Proposition 4** (Finite class). *Suppose $\Theta$ is a finite set ($N_{\Theta} := |\Theta| < \infty$), and suppose Assumptions A1, A2, B hold. Then, we have with probability at least $1 - \delta$ that*

$$\varepsilon_{\text{coverage}} \leq C\sqrt{\log(N_{\Theta}/\delta)/n_{\text{cal}}} \quad \text{and} \quad \varepsilon_{\text{eff}} \leq C \cdot \left[M\sqrt{\log(N_{\Theta}/\delta)} + L_{\mathcal{T}}B_{\mathcal{T}}\right]/\sqrt{n_{\text{cal}}},$$

*where $C > 0$ is an absolute constant.*

*Proof.* We first bound $\varepsilon_{\text{coverage}}$. Fix any $\theta \in \Theta$, define

$$t_{\theta}(x,y) := \inf\{t \in \mathcal{T} : C_{\theta,t}(x) \ni y\} \tag{10}$$

to be the smallest possible $t \in \mathcal{T}$ such that the $C_{\theta,t}(x)$ contains $y$. Observe that, as $\{C_{\theta,t}(x)\}_{t\in\mathcal{T}}$ are nested sets, the coverage event can be rewritten as $\mathbf{1}\{y \in C_{\theta,t}(x)\} = \mathbf{1}\{t \geq t_{\theta}(x,y)\}$ for any $(x,y)$. Therefore, we have

$$\sup_{t\in\mathcal{T}} \left|\widehat{L}_{\text{coverage}}(C_{\theta,t}) - L_{\text{coverage}}(C_{\theta,t})\right|$$

$$= \sup_{t\in\mathcal{T}} \left|\frac{1}{n_{\text{cal}}} \sum_{i=1}^{n_{\text{cal}}} \mathbf{1}\{y_i \notin C_{\theta,t}(x_i)\} - \mathbb{P}(Y \notin C_{\theta,t}(X))\right|$$

$$= \sup_{t\in\mathcal{T}} \left|\frac{1}{n_{\text{cal}}} \sum_{i=1}^{n_{\text{cal}}} \mathbf{1}\{y_i \in C_{\theta,t}(x_i)\} - \mathbb{P}(Y \in C_{\theta,t}(X))\right|$$

$$= \sup_{t\in\mathcal{T}} \left|\frac{1}{n_{\text{cal}}} \sum_{i=1}^{n_{\text{cal}}} \mathbf{1}\{t_{\theta}(x,y) \leq t\} - \mathbb{P}_{(x,y)}(t_{\theta}(X,Y) \leq t)\right|$$

$$= \sup_{t\in\mathcal{T}} \left|\widehat{F}_{\theta}(t) - F_{\theta}(t)\right|,$$

where we have defined $F_{\theta} : \mathbb{R} \to [0,1]$ as the CDF of the random variable $t_{\theta}(X,Y)$ and similarly $\widehat{F}_{\theta}$ as the empirical CDF of the same random variable over the finite dataset $D_{\text{cal}}$. Applying the Dvoretzky-Kiefer-Wolfowitz (DKW) inequality (Massart, 1990, Corollary 1) yields that

$$\sup_{t\in\mathcal{T}} \left|\widehat{F}_{\theta}(t) - F_{\theta}(t)\right| \leq \sqrt{\frac{\log(2/\delta)}{2n_{\text{cal}}}}$$

with probability at least $1 - \delta$. Now, taking the union bound with respect to $\theta \in \Theta$ (where for each $\theta$ we plug in tail probability $\delta/2N_\Theta$) we get that with probability at least $1 - \delta/2$,

$$
\begin{aligned}
\varepsilon_{\text{coverage}} &= \sup_{\theta \in \Theta} \sup_{t \in \mathcal{T}} \left| \widehat{L}_{\text{coverage}}(C_{\theta,t}) - L_{\text{coverage}}(C_{\theta,t}) \right| \\
&= \sup_{\theta \in \Theta} \sup_{t \in \mathcal{T}} \left| \widehat{F}_\theta(t) - F_\theta(t) \right| \leq \sqrt{\frac{\log(4N_\Theta/\delta)}{2n_{\text{cal}}}} \leq C\sqrt{\frac{\log(N_\Theta/\delta)}{n_{\text{cal}}}}.
\end{aligned}
\tag{11}
$$

for some absolute constant $C > 0$.

We next bound $\varepsilon_{\text{eff}}$. Fix any $\theta \in \Theta$. We have by standard symmetrization argument that

$$
\begin{aligned}
&\mathbb{E}\left[\sup_{t \in \mathcal{T}} \left| \widehat{L}_{\text{eff}}(C_{\theta,t}) - L_{\text{eff}}(C_{\theta,t}) \right|\right] \\
&= \mathbb{E}\left[\sup_{t \in \mathcal{T}} \left| \frac{1}{n_{\text{cal}}} \sum_{i=1}^{n_{\text{cal}}} \ell_{\text{eff}}(C_{\theta,t}; (x_i, y_i)) - \mathbb{E}[\ell_{\text{eff}}(C_{\theta,t}; (X, Y))] \right|\right] \\
&\leq 2\mathbb{E}_{(x_i,y_i),\varepsilon_i}\left[\sup_{t \in \mathcal{T}} \left| \frac{1}{n_{\text{cal}}} \sum_{i=1}^{n_{\text{cal}}} \varepsilon_i \ell_{\text{eff}}(C_{\theta,t}; (x_i, y_i)) \right|\right] \\
&\overset{(i)}{\leq} 2L_{\mathcal{T}} \cdot \mathbb{E}_{\varepsilon_i}\left[\sup_{t \in \mathcal{T}} \left| \frac{1}{n_{\text{cal}}} \sum_{i=1}^{n_{\text{cal}}} \varepsilon_i \cdot t \right|\right] = 2L_{\mathcal{T}} \cdot \mathbb{E}_{\varepsilon_i}\left[\left| \frac{1}{n_{\text{cal}}} \sum_{i=1}^{n_{\text{cal}}} \varepsilon_i \right| \cdot \sup_{t \in \mathcal{T}} |t|\right] \\
&\overset{(ii)}{\leq} 2L_{\mathcal{T}} \cdot B_{\mathcal{T}} \cdot \mathbb{E}_{\varepsilon_i}\left[\left| \frac{1}{n_{\text{cal}}} \sum_{i=1}^{n_{\text{cal}}} \varepsilon_i \right|\right] \overset{(iii)}{\leq} 2L_{\mathcal{T}} \cdot B_{\mathcal{T}}/\sqrt{n_{\text{cal}}}.
\end{aligned}
$$

Above, (i) used the Lipschitzness Assumption A2 and the Rademacher contraction inequality (Vershynin, 2018, Exercise 6.7.7); (ii) used Assumption B, and (iii) used $\mathbb{E}_{\varepsilon_i}\left[\left| \frac{1}{n_{\text{cal}}} \sum_{i=1}^{n_{\text{cal}}} \varepsilon_i \right|\right] \leq \left(\mathbb{E}_{\varepsilon_i}\left[\left(\frac{1}{n_{\text{cal}}} \sum_{i=1}^{n_{\text{cal}}} \varepsilon_i\right)^2\right]\right)^{1/2} = 1/\sqrt{n_{\text{cal}}}$. (Above $\varepsilon_i \overset{\text{iid}}{\sim} \text{Unif}(\{\pm 1\})$ are Rademacher variables.)

Next, as each loss $|\ell_{\text{eff}}(C_{\theta,t}; (x, y))| \leq M$ by Assumption A1, the random variable

$$
\sup_{t \in \mathcal{T}} \left| \widehat{L}_{\text{eff}}(C_{\theta,t}) - L_{\text{eff}}(C_{\theta,t}) \right|
$$

satisfies the $M/n_{\text{cal}}$ finite-difference property. Therefore by McDiarmid's Inequality, we have with probability at least $1 - \delta$ that

$$
\begin{aligned}
&\sup_{t \in \mathcal{T}} \left| \widehat{L}_{\text{eff}}(C_{\theta,t}) - L_{\text{eff}}(C_{\theta,t}) \right| \\
&\leq \mathbb{E}\left[\sup_{t \in \mathcal{T}} \left| \widehat{L}_{\text{eff}}(C_{\theta,t}) - L_{\text{eff}}(C_{\theta,t}) \right|\right] + \sqrt{\frac{M^2 \log(1/\delta)}{2n_{\text{cal}}}} \\
&\leq C \cdot \frac{L_{\mathcal{T}} B_{\mathcal{T}} + M\sqrt{\log(1/\delta)}}{\sqrt{n_{\text{cal}}}}.
\end{aligned}
$$

Finally, by union bound over $\theta \in \Theta$ (where we plug in $\delta/2N_\Theta$ as tail probability into the above), we have with probability at least $1 - \delta/2$ that

$$
\begin{aligned}
\varepsilon_{\text{eff}} &= \sup_{\theta \in \Theta} \sup_{t \in \mathcal{T}} \left| \widehat{L}_{\text{eff}}(C_{\theta,t}) - L_{\text{eff}}(C_{\theta,t}) \right| \\
&\leq C \cdot \frac{L_{\mathcal{T}} B_{\mathcal{T}} + M\sqrt{\log(N_\Theta/\delta)}}{\sqrt{n_{\text{cal}}}}.
\end{aligned}
\tag{12}
$$

(11) together with (12) is the desired result. $\qquad\square$

## C.2 VC/RADEMACHER CLASS

Next, for any class $\mathcal{C}$, let $\mathrm{VC}(\mathcal{C}) := \mathrm{VC}(\{(x,y) \mapsto \mathbf{1}\{y \notin C_{\theta,t}(x)\} : \theta \in \Theta, t \in \mathcal{T}\})$ denote its VC dimension with respect to the coverage loss.

**Proposition 5** (VC/Rademacher class). *We have for some absolute constant $C > 0$ that*

*(a) Suppose $\mathrm{VC}(\mathcal{C}) = K + 1 < \infty$, then with probability at least $1 - \delta/2$,*

$$\varepsilon_{\mathrm{coverage}} \le C\sqrt{(K + 1 + \log(1/\delta))/n_{\mathrm{cal}}}.$$

*(b) Suppose Assumption A1 holds. Then we have with probability at least $1 - \delta/2$ that*

$$\varepsilon_{\mathrm{eff}} \le C\Big[R_{n_{\mathrm{cal}}}^{\mathrm{eff}}(\mathcal{C}) + \sqrt{M^2 \log(1/\delta)/n_{\mathrm{cal}}}\Big],$$

*where $R_{n_{\mathrm{cal}}}^{\mathrm{eff}}(\mathcal{C}) := \mathbb{E}_{(x_i,y_i),\varepsilon_i}\Big[\sup_{(\theta,t) \in \Theta \times \mathcal{T}} \Big|\frac{1}{n_{\mathrm{cal}}} \sum_{i=1}^{n_{\mathrm{cal}}} \varepsilon_i \ell_{\mathrm{eff}}(C_{\theta,t}; (x_i, y_i))\Big|\Big]$ is the Rademacher complexity of the class $\mathcal{C}$ with respect to $\ell_{\mathrm{eff}}$ (above $\varepsilon_i \overset{\mathrm{iid}}{\sim} \mathrm{Unif}(\{\pm 1\})$).*

*Proof.* (a) By assumption, the class of Boolean functions $\{(x,y) \mapsto \mathbf{1}\{y \notin C_{\theta,t}(x)\}\}_{(\theta,t) \in \Theta \times \mathcal{T}}$ has VC dimension $K + 1 < \infty$. Therefore by the standard Rademacher complexity bound for VC classes (Vershynin, 2018, Theorem 8.3.23) and McDiarmid's Inequality, we have with probability at least $1 - \delta/2$ that

$$\varepsilon_{\mathrm{coverage}} = \sup_{(\theta,t) \in \Theta \times \mathcal{T}} \left|\frac{1}{n_{\mathrm{cal}}} \sum_{i=1}^{n_{\mathrm{cal}}} \mathbf{1}\{y_i \notin C_{\theta,t}(x_i)y_i\} - \mathbb{P}(Y \notin C_{\theta,t}(X))\right|$$

$$\le C\sqrt{\frac{K+1}{n_{\mathrm{cal}}}} + \sqrt{\frac{\log(2/\delta)}{2n_{\mathrm{cal}}}} \le C\sqrt{\frac{K + 1 + \log(1/\delta)}{n_{\mathrm{cal}}}}.$$

(b) We have by standard symmetrization argument that (below $\varepsilon_i \overset{\mathrm{iid}}{\sim} \mathrm{Unif}(\{\pm 1\})$ denote Rademacher variables)

$$\mathbb{E}[\varepsilon_{\mathrm{eff}}] = \mathbb{E}\left[\sup_{\theta \in \Theta, t \in \mathcal{T}} \left|\widehat{L}_{\mathrm{eff}}(C_{\theta,t}) - L_{\mathrm{eff}}(C_{\theta,t})\right|\right]$$

$$= \mathbb{E}\left[\sup_{\theta \in \Theta, t \in \mathcal{T}} \left|\frac{1}{n_{\mathrm{cal}}} \sum_{i=1}^{n_{\mathrm{cal}}} \ell_{\mathrm{eff}}(C_{\theta,t}; (x_i, y_i)) - \mathbb{E}[\ell_{\mathrm{eff}}(C_{\theta,t}; (X, Y))]\right|\right]$$

$$\le 2\mathbb{E}_{(x_i,y_i),\varepsilon_i}\left[\sup_{\theta \in \Theta, t \in \mathcal{T}} \left|\frac{1}{n_{\mathrm{cal}}} \sum_{i=1}^{n_{\mathrm{cal}}} \varepsilon_i \ell_{\mathrm{eff}}(C_{\theta,t}; (x_i, y_i))\right|\right] = 2R_n(\mathcal{C}).$$

Further by Assumption A1, the quantity $\varepsilon_{\mathrm{eff}}$ satisfies $M/n_{\mathrm{cal}}$ bounded-difference, so applying McDiarmid's Inequality gives that with probability at least $1 - \delta/2$,

$$\varepsilon_{\mathrm{eff}} \le \mathbb{E}[\varepsilon_{\mathrm{eff}}] + \sqrt{\frac{2M^2 \log(2/\delta)}{n_{\mathrm{cal}}}} \le C\left[R_{n_{\mathrm{cal}}}^{\mathrm{eff}}(\mathcal{C}) + \sqrt{\frac{M^2 \log(1/\delta)}{n}}\right].$$

$\square$

## C.3 CASE STUDY: LINEAR CLASS

In this section, we study prediction intervals with a specific linear structure and show that it satisfies the conditions of the VC/Rademacher class of Proposition 5.

Concretely, suppose we have a regression task ($\mathcal{Y} = \mathbb{R}$), and the prediction interval $C_{\theta,t}(x)$ takes a linear form

$$C_{\theta,t}(x) = [\theta^\top \Phi_{\mathrm{lo}}(x) - t\sigma(x), \theta^\top \Phi_{\mathrm{hi}}(x) + t\sigma(x)], \tag{13}$$

where $\theta \in \Theta \subset \mathbb{R}^K$, $\Phi_{\text{hi}}, \Phi_{\text{lo}} : \mathcal{X} \to \mathbb{R}^K$ are feature maps such that $\Phi_{\text{lo}}(x)_i \leq \Phi_{\text{hi}}(x)_i$ for all $i \in [K]$, $\sigma : \mathcal{X} \to \mathbb{R}_{>0}$.

For intuitions, we can think of $\Phi_{\{\text{hi,lo}\}}$ as pretrained representation functions and $\sigma$ as an (optional) pretrained function for modeling the variability of $y|x$. Note that this encompasses linear ensembling of several existing methods, such as vanilla conformal regression (Lei et al., 2018) by taking $\Phi_{\text{hi}} = \Phi_{\text{lo}} = \Phi$ where each $\Phi_i : \mathcal{X} \to \mathbb{R}$ is a base predictor, as well as Conformalized Quantile Regression (Romano et al., 2019) where each $(\Phi_{\text{lo},i}, \Phi_{\text{hi},i})$ is a pair of learned lower and upper quantile functions.

Our goal is to find an optimal linear function of this representation that yields the shortest prediction interval (with fixed width) subject to valid coverage.

We assume that both the features and the parameters are bounded:

**Assumption C** (Bounded features and parameters). *We have* $\sup_{\theta \in \Theta} \|\theta\| \leq B_\Theta$, $\sup_{x \in \mathcal{X}} \|\Phi(x)\| \leq B_\Phi$, $\sup_{x \in \mathcal{X}} \sigma(x) \leq B_\sigma$, *and* $\sup_{t \in \mathcal{T}} |t| \leq B_\mathcal{T}$.

The following result shows that Proposition 5 is applicable on the linear class.

**Corollary 6** (Coverage and length guarantees for linear class). *For the $(K+1)$-dimensional linear class (13), suppose Assumption C holds, and we take the efficiency loss to be the length of the interval: $\ell_{\text{eff}}(C; (x,y)) := \text{length}(C(x))$. Then, we have with probability at least $1 - \delta$ (over the calibration dataset $D_{\text{cal}}$) that*

$$\varepsilon_{\text{coverage}} \leq C\sqrt{\frac{K+1+\log(1/\delta)}{n_{\text{cal}}}}, \quad \text{and} \quad \varepsilon_{\text{eff}} \leq C[B_\Theta B_\Phi + B_\mathcal{T} B_\sigma] \cdot \sqrt{\frac{\log(1/\delta)}{n_{\text{cal}}}},$$

*where $C > 0$ is an absolute constant.*

*Proof.* We verify the conditions of Proposition 5. First, we have

$$\mathbf{1}\{y \notin C_{\theta,t}(x)\} = \mathbf{1}\left\{\max\left\{y - \theta^\top \Phi_{\text{hi}}(x), \theta^\top \Phi_{\text{lo}}(x) - y\right\} > t\sigma(x)\right\}.$$

The set within the indicator above is the union of two sets $\{(x,y) : y - \theta^\top \Phi_{\text{hi}}(x) - t\sigma(x) > 0\}$ and $\{(x,y) : \theta^\top \Phi_{\text{lo}}(x) - y - t\sigma(x) > 0\}$. Note that each family of sets (over $(\theta, t) \in \mathbb{R}^K \times \mathbb{R}$ are linear halfspaces with feature dimension $K + 2$), and thus has VC-dimension $\leq K + 2$. Applying the VC dimension bound for unions of sets (Van Der Vaart & Wellner, 2009, Theorem 1.1), we get $\text{VC}(\mathcal{C}) \leq C'(K + 2 + K + 2) \leq C(K + 1)$ for some absolute constant $C > 0$. Therefore the condition of Proposition 5(a) holds from which we obtain the desired bound for $\varepsilon_{\text{coverage}}$.

To bound $\varepsilon_{\text{eff}}$, we first note that for any $(x, y) \in \mathcal{X} \times \mathbb{R}$,

$$\begin{aligned} |\ell_{\text{eff}}(C_{\theta,t}; (x,y))| &= |\text{length}(C_{\theta,t}(x))| \\ &= \theta^\top(\Phi_{\text{hi}}(x) - \Phi_{\text{lo}}(x)) + 2t\sigma(x) \leq \|\theta\| \|\Phi_{\text{hi}}(x) - \Phi_{\text{lo}}(x)\| + 2t\sigma(x) \\ &\leq 2B_\Theta B_\Phi + 2B_\mathcal{T} B_\sigma =: M, \end{aligned}$$

and thus the boundedness assumption (Assumption A1) holds with $M$ defined above. Next, we have the following bound on the Rademacher complexity

$$R_{n_{\text{cal}}}^{\text{eff}}(\mathcal{C}) = \mathbb{E}\left[\sup_{(\theta,t) \in \Theta \times \mathcal{T}} \left|\frac{1}{n_{\text{cal}}} \sum_{i=1}^{n_{\text{cal}}} \varepsilon_i\left(\theta^\top(\Phi_{\text{hi}}(x_i) - \Phi_{\text{lo}}(x_i)) + 2t\sigma(x_i)\right)\right|\right]$$

$$\leq \mathbb{E}\left[\sup_{\theta \in \Theta} \left|\left\langle \theta, \frac{1}{n_{\text{cal}}} \sum_{i=1}^{n_{\text{cal}}} \varepsilon_i(\Phi_{\text{hi}}(x_i) - \Phi_{\text{lo}}(x_i))\right\rangle\right|\right] + \mathbb{E}\left[\sup_{t \in \mathcal{T}} \left|2t \cdot \frac{1}{n_{\text{cal}}} \sum_{i=1}^{n_{\text{cal}}} \varepsilon_i \sigma(x_i)\right|\right]$$

$$\leq \sup_{\theta \in \Theta} \|\theta\| \cdot \mathbb{E}\left[\left\|\frac{1}{n_{\text{cal}}} \sum_{i=1}^{n_{\text{cal}}} \varepsilon_i(\Phi_{\text{hi}}(x_i) - \Phi_{\text{lo}}(x_i))\right\|^2\right]^{1/2} + 2\sup_{t \in \mathcal{T}} |t| \cdot \mathbb{E}\left[\left(\frac{1}{n_{\text{cal}}} \sum_{i=1}^{n_{\text{cal}}} \varepsilon_i \sigma(x_i)\right)^2\right]^{1/2}$$

$$\leq B_\Theta \cdot \mathbb{E}\left[\frac{1}{n_{\text{cal}}} \|\Phi_{\text{hi}}(x_1) - \Phi_{\text{lo}}(x_1)\|^2\right]^{1/2} + 2B_\mathcal{T} \cdot \mathbb{E}\left[\frac{1}{n_{\text{cal}}} \sigma^2(x_1)\right]^{1/2}$$

$$\leq C \cdot \frac{B_\Theta B_\Phi + B_\mathcal{T} B_\sigma}{\sqrt{n_{\text{cal}}}}.$$

Applying Proposition 5(b), we get $\varepsilon_{\text{eff}} \leq C \cdot [B_\Theta B_\Phi + B_\mathcal{T} B_\sigma] \cdot \sqrt{\log(1/\delta)/n_{\text{cal}}}$ with probability at least $1 - \delta$. This is the desired bound for $\varepsilon_{\text{eff}}$. $\qquad\square$

## D    THEORETICAL GUARANTEE FOR CP-Gen-Recal

In this section we state and prove the formal theoretical guarantee for the CP-Gen-Recal algorithm (Algorithm 2).

Define the score $t_\theta(X, Y) := \inf \{t \in \mathcal{T} : Y \in C_{\theta,t}(X)\}$ and let $F_\theta(t) := \mathbb{P}(Y \in C_{\theta,t}(X)) = \mathbb{P}(t_\theta(X, Y) \leq t)$ denote its CDF.

**Assumption D** (Lower bounded density for score function). *For any $\theta \in \Theta$, $t_\theta(X, Y)$ has a positive density $f_\theta(t) = F'_\theta(t) > 0$ on $t \in \mathcal{T}$. Further, let $t_{\theta,1-\alpha} := \inf \{t \in \mathcal{T} : F_\theta(t) \geq 1 - \alpha\}$ denote its $(1 - \alpha)$ quantile, then there exists some constants $\underline{c}_0, \delta_0 > 0$ such that*

$$\inf_{t \in [t_{\theta,1-\alpha}-\delta_0, t_{\theta,1-\alpha}+\delta_0]} f_\theta(t) \geq \underline{c}_0.$$

**Proposition 7** (Valid coverage and near-optimal efficiency for reconformalized algorithm). *The following holds for Algorithm 2:*

*(a) (Valid coverage) For any possible $\widehat{\theta} \in \Theta$ learned in Line 1 and the resulting $\widehat{t}_{\text{recal}}$, we have*

$$\mathbb{E}_{D_{\text{recal}}} \left[ L_{\text{coverage}}(C_{\widehat{\theta},\widehat{t}_{\text{recal}}}) \right] \leq \alpha, \quad \text{and thus} \quad \mathbb{P}_{D_{\text{recal}},(X,Y)} \left( Y \in C_{\widehat{\theta},\widehat{t}_{\text{recal}}}(X) \right) \geq 1 - \alpha.$$

*(b) (Efficiency) Suppose Assumptions A2 and D hold, $\max \left\{ \varepsilon_{\text{coverage}} + 1/n_{\text{cal}}, 2\sqrt{\log(1/\delta)/n_{\text{recal}}} \right\} \leq \underline{c}_0 \delta_0$ (recall the definition of $\varepsilon_{\text{coverage}}$ in (7)), and $\varepsilon_{\text{coverage}} \leq \varepsilon_0$. Then for $\delta \leq 0.5$, we have with probability at least $1 - \delta$ that*

$$L_{\text{eff}}(C_{\widehat{\theta},\widehat{t}_{\text{recal}}}) \leq \min_{\substack{(\theta,t) \in \Theta \times \mathcal{T} \\ L_{\text{coverage}}(C_{\theta,t}) \leq \alpha}} L_{\text{eff}}(C_{\theta,t}) + 2\varepsilon_{\text{eff}} + CL_\mathcal{T} \cdot \left[ \varepsilon_{\text{coverage}} + \frac{1}{n_{\text{cal}}} + \sqrt{\frac{\log(1/\delta)}{n_{\text{recal}}}} \right] / \underline{c}_0.$$

*Proof.* (a) As the learned parameter $\widehat{\theta}$ (and thus the family of nested sets $C_{\widehat{\theta},t}$) is independent of the recalibration dataset $D_{\text{recal}}$, we have that the scores $t_{\widehat{\theta}}(x, y)$ on dataset $D_{\text{recal}}$ and a new test point $(X, Y)$ are exchangeable given any $\widehat{\theta}$. Therefore by (Gupta et al., 2019, Proposition 1), we have for any $\widehat{\theta} \in \Theta$ that

$$\mathbb{P}_{D_{\text{recal}},(X,Y)} \left( Y \in C_{\widehat{\theta},\widehat{t}_{\text{recal}}}(X) \right) \geq 1 - \alpha,$$

or equivalently $\mathbb{E}_{D_{\text{recal}}} \left[ L_{\text{coverage}}(C_{\widehat{\theta},\widehat{t}_{\text{recal}}}) \right] \leq \alpha$.

(b) For any $\theta \in \Theta$, define the score function $t_\theta(x, y)$ the same as in (10), and similarly define the CDF $F_\theta(t) := \mathbb{P}(t_\theta(X, Y) \leq t)$ and its empirical counterpart $\widehat{F}_\theta^{\text{cal}}(t)$ and $\widehat{F}_\theta^{\text{recal}}(t)$ as the finite-sample version on dataset $D_{\text{cal}}$ and $D_{\text{recal}}$ respectively.

We first analyze $\widehat{t}$. By the same derivation as in (11), we have

$$\sup_{t \in \mathcal{T}} \left| \widehat{F}_{\widehat{\theta}}^{\text{cal}}(t) - F_{\widehat{\theta}}(t) \right| \leq \sup_{(\theta,t) \in \Theta \times \mathcal{T}} \left| \widehat{F}_\theta^{\text{cal}}(t) - F_\theta(t) \right| = \varepsilon_{\text{coverage}}.$$

As $(\widehat{\theta}, \widehat{t})$ solves the constrained ERM (5) and by the assumption that $\ell_{\text{eff}}(C_{\theta,t}; (x, y))$ is monotone in $t$, we have that $\widehat{t}$ is the minimal value of $t \in \mathcal{T}$ such that $\widehat{F}_{\widehat{\theta}}^{\text{cal}}(t) \geq 1 - \alpha$. Therefore, (as $|D_{\text{cal}}| = n_{\text{cal}}$ and $\{t_\theta(x_i, y_i)\}_{i \in D_{\text{cal}}}$ are almost surely distinct by Assumption D,) we have

$$1 - \alpha \leq \widehat{F}_{\widehat{\theta}}^{\text{cal}}(\widehat{t}) \leq 1 - \alpha + 1/n_{\text{cal}}.$$

This shows that

$$\left|F_{\widehat{\theta}}(\widehat{t}) - F_{\widehat{\theta}}(t_{\widehat{\theta},1-\alpha})\right| = \left|F_{\widehat{\theta}}(\widehat{t}) - (1-\alpha)\right| \le \varepsilon_{\text{coverage}} + 1/n_{\text{cal}},$$

where we recall that $t_{\widehat{\theta},1-\alpha}$ is the $(1-\alpha)$ (population) quantile of $t_{\widehat{\theta}}(X,Y)$. Note that $F'_{\theta}(t) = f_{\theta}(t) \ge \underline{c}_0$ on $t \in [t_{\widehat{\theta},1-\alpha} - \delta_0, t_{\widehat{\theta},1-\alpha} + \delta_0]$ by Assumption D. Further, $\varepsilon_{\text{coverage}} + 1/n_{\text{cal}} \le \underline{c}_0\delta_0$. Therefore, by monotonicity of $F_{\theta}$, we must have $\widehat{t} \in [t_{\widehat{\theta},1-\alpha} - \delta_0, t_{\widehat{\theta},1-\alpha} + \delta_0]$, and thus

$$\left|\widehat{t} - t_{\widehat{\theta},1-\alpha}\right| \le (\varepsilon_{\text{coverage}} + 1/n_{\text{cal}})/\underline{c}_0. \tag{14}$$

We next analyze $\widehat{t}_{\text{recal}}$. As the dataset $D_{\text{recal}}$ is independent of $\widehat{\theta}$, we can apply the DKW Inequality (Massart, 1990, Corollary 1) to obtain that

$$\sup_{t \in \mathcal{T}} \left|\widehat{F}_{\widehat{\theta}}^{\text{recal}}(t) - F_{\widehat{\theta}}(t)\right| \le \sqrt{\frac{\log(1/\delta)}{2n_{\text{recal}}}}$$

with probability at least $1-\delta$. Using a similar argument as above, we get (for $\delta \le 0.5$)

$$\left|F_{\widehat{\theta}}(\widehat{t}_{\text{recal}}) - F_{\widehat{\theta}}(t_{\widehat{\theta},1-\alpha})\right| \le \sqrt{\frac{\log(1/\delta)}{2n_{\text{recal}}}} + \frac{1}{n_{\text{recal}}} \le 2\sqrt{\frac{\log(1/\delta)}{n_{\text{recal}}}}.$$

As $2\sqrt{\log(1/\delta)/n_{\text{recal}}} \le \underline{c}_0\delta_0$, we can apply the similar argument as above to deduce that

$$\left|\widehat{t}_{\text{recal}} - t_{\widehat{\theta},1-\alpha}\right| \le 2\sqrt{\log(1/\delta)/n_{\text{recal}}}/\underline{c}_0. \tag{15}$$

Combining (14) and (15) and using the Lipschitzness of the efficiency loss (Assumption A2), we get

$$L_{\text{eff}}(C_{\widehat{\theta},\widehat{t}_{\text{recal}}}) - L_{\text{eff}}(C_{\widehat{\theta},\widehat{t}})$$
$$\le L_{\mathcal{T}} \cdot \left|\widehat{t}_{\text{recal}} - \widehat{t}\right| \le L_{\mathcal{T}} \cdot \left(\left|\widehat{t}_{\text{recal}} - t_{\widehat{\theta},1-\alpha}\right| + \left|t_{\widehat{\theta},1-\alpha} - \widehat{t}\right|\right)$$
$$\le CL_{\mathcal{T}} \cdot \left[\varepsilon_{\text{coverage}} + n_{\text{cal}}^{-1} + \sqrt{\frac{\log(1/\delta)}{n_{\text{recal}}}}\right]/\underline{c}_0.$$

Finally, as we assumed $\varepsilon_{\text{coverage}} \le \varepsilon_0$, the condition of Proposition 2(b) holds, so we have

$$L_{\text{eff}}(C_{\widehat{\theta},\widehat{t}}) \le \inf_{\substack{(\theta,t) \in \Theta \times \mathcal{T} \\ L_{\text{coverage}}(C_{\theta,t}) \le \alpha}} L_{\text{eff}}(C_{\theta,t}) + 2\varepsilon_{\text{eff}}.$$

Summing the preceding two bounds, we get

$$L_{\text{eff}}(C_{\widehat{\theta},\widehat{t}_{\text{recal}}}) \le \inf_{\substack{(\theta,t) \in \Theta \times \mathcal{T} \\ L_{\text{coverage}}(C_{\theta,t}) \le \alpha}} L_{\text{eff}}(C_{\theta,t}) + 2\varepsilon_{\text{eff}} + CL_{\mathcal{T}} \cdot \left[\varepsilon_{\text{coverage}} + n_{\text{cal}}^{-1} + \sqrt{\frac{\log(1/\delta)}{n_{\text{recal}}}}\right]/\underline{c}_0.$$

which is the desired result.

$\square$

# E    ADDITIONAL EXPERIMENTAL DETAILS

## E.1    CONFORMAL QUANTILE FINETUNING

**Datasets**    Our choice of the datasets follows (Feldman et al., 2021). We provide information about these datasets in Table 3.

All datasets are standardized so that inputs and labels have mean 0 and standard deviation 1, and split into (train, cal, recal, test) with size 70%, 10%, 10%, 10% (varying with the random seed).

Table 3: Information about the regression datasets. Here $(n, d)$ denotes the (sample size, feature dim).

| Dataset | $n$ | $d$ |
|---|---|---|
| MEPS_19 (mep, a) | 15785 | 139 |
| MEPS_20 (mep, b) | 17541 | 139 |
| MEPS_21 (mep, c) | 15656 | 139 |
| Facebook_1 (fac) | 40948 | 53 |
| Facebook_2 (fac) | 81311 | 53 |
| kin8nm (kin) | 8192 | 8 |
| naval (nav) | 11934 | 17 |
| bio (bio) | 45730 | 9 |
| blog_data (blo) | 52397 | 280 |

**Base predictor and optimization**  Our network architecture is a 3-layer MLP with width 64 and output dimension 2 (for the lower and upper quantile). We use momentum SGD with initial learning rate $10^{-3}$ and momentum 0.9, batch-size 1024, and run the optimization for a max of 10000 epochs. A 10x learning rate decay is performed if the validation loss on $D_{\text{cal}}$ has not decreased in 10 epochs, and we stop the learning whenever the learning rate decay happens for 3 times. The loss function used in training $\widehat{F} = [\widehat{f}_{\text{lo}}, \widehat{f}_{\text{hi}}]$ is the summed pinball loss of level $\alpha/2$ for $\widehat{f}_{\text{lo}}$ and $1 - \alpha/2$ for $\widehat{f}_{\text{hi}}$, following (Romano et al., 2019):

$$\ell(\widehat{F}; (x_i, y_i)) = \ell_{\text{pinball}}^{\alpha/2}(\widehat{f}_{\text{lo}}(x_i) - y_i) + \ell_{\text{pinball}}^{1-\alpha/2}(\widehat{f}_{\text{hi}}(x_i) - y_i),$$

where for any $\beta \in (0, 1)$, $\ell_{\text{pinball}}^\beta$ is the pinball loss at level $\beta$:

$$\ell_{\text{pinball}}^\beta(t) = \begin{cases} -\beta t & \text{if } t < 0, \\ (1 - \beta)t & \text{if } t \geq 0. \end{cases}$$

**Optimization details for** `CP-Gen-Recal`  For the conformal quantile finetuning procedure with our `CP-Gen-Recal`, we rewrite the miscoverage loss for the quantile-based prediction interval as

$$\mathbf{1}\left\{y \notin C_{\theta,t}(x)\right\} = \mathbf{1}\left\{t - \max\left\{\theta_{\text{lo}}^\top \widehat{\Phi}(x) - y, y - \theta_{\text{hi}}^\top \widehat{\Phi}(x)\right\} < 0\right\}.$$

(In practice our $\theta$ also includes a trainable bias same as the original top linear layer; here we abuse notation slightly to allow easier presentation.) We approximate the right-hand side above with the hinge loss to obtain the formulation (8). To solve that optimization problem, we use SGD on $(\theta, t)$ with learning rate 0.01 and (ascent on) $\lambda$ with learning rate 0.1. The batch-size here is 256 and the number of episodes is 1000. To ensure $t > 0$ we use a log parametrization for $t$. Finally, $t_{\text{recal}}$ is computed by the reconformalization step in Algorithm 2 on $D_{\text{recal}}$.

### E.2  Multi-output regression

**Datasets**  We generate offline datasets consisting of (state, action, next_state) pairs within RL tasks within the OpenAI Gym (Brockman et al., 2016). For each task, the data is generated by executing a medium-performing *behavior policy* that is extracted from standard RL training runs. All tasks are continuous state and continuous action. Table 4 summarizes the state and action dimension, along with the reward of the policies used for generating the data. All datasets contain 200K examples.

All datasets are standardized so that inputs and labels have mean 0 and standard deviation 1, and split into (train, cal, recal, test) with size 70%, 10%, 10%, 10% (varying with the random seed).

**Base predictor and optimization**  Our network architecture is a 3-layer MLP with width 64, input dimension $d_{\text{in}} = d_S + d_A$, and output dimension $d_{\text{out}} = d_S$. We use momentum SGD with initial learning rate $10^{-3}$, momentum 0.9, and batch-size 512. We run the optimization for 1000 epochs with a 10x learning rate decay at epoch 500. The loss function for training the network is the standard MSE loss.

**Optimization details for** `CP-Gen-Recal`  For the conformal quantile finetuning procedure with our `CP-Gen-Recal`, we rewrite the miscoverage loss for the box-shaped prediction set as

$$\mathbf{1}\left\{y \notin C_u(x)\right\} = \mathbf{1}\left\{y \notin \prod_{i=1}^{d_{\text{out}}}[\widehat{f}_i(x) - u_i, \widehat{f}_i(x) + u_i]\right\} = \mathbf{1}\left\{1 - \max_{1 \leq i \leq d_{\text{out}}} |y_i - \widehat{f}_i(x)|/u_i < 0\right\}.$$

Table 4: Information about the next-state prediction datasets. Here $(d_S, d_A)$ denotes the (state, action) dimension of the corresponding RL task. Datasets with a (slim) note only extract a subset of the full state (so that $d_S$ is less than the full state dimension). We also report the mean reward of the behavior policies.

| RL Task | $d_S$ | $d_A$ | mean reward |
|---|---|---|---|
| Cartpole | 4 | 1 | 107 |
| Half-Cheetah | 17 | 6 | 8015 |
| Ant (slim) | 27 | 8 | 4645 |
| Walker | 17 | 6 | 3170 |
| Swimmer | 8 | 2 | 51 |
| Hopper | 11 | 3 | 2066 |
| Humanoid (slim) | 45 | 17 | 1357 |

where we recall $u \in \mathbb{R}^{d_{\text{out}}}$ is the learnable parameter within the initial optimization stage of CP-Gen-Recal as discussed in Section 5.2. We approximate the right-hand side above with the hinge loss to obtain the formulation (8). To solve that optimization problem, we use SGD on $(\theta, t)$ with learning rate 0.01 and (ascent on) $\lambda$ with learning rate 0.01. The batch-size here is 1024 and the number of episodes is 1000. To ensure $u > 0$ we use a log parametrization for $u$.

For the reconformalization step, we keep the (relative) ratios of the $\widehat{u}$ obtained above (as the $\widehat{\theta}$), and then reconformalize an additional $t_{\text{recal}} > 0$ on $D_{\text{recal}}$ via the proportional reshaping of (9).

### E.3 DETAILS FOR FIGURE 1

Figure 1 is obtained on one run of our conformal quantile finetuning experiment on the MEPS_19 dataset, and illustrates the coverage-efficiency tradeoff. Both figures there compute the coverage and length on the (unseen) test set $D_{\text{test}}$, for better illustration. Figure 1 Left plots the family $[\widehat{f}_{\text{lo}}(x) - t, \widehat{f}_{\text{hi}}(x) + t]$ used by Conformalized Quantile Regression. Figure 1 Right plots the family

$$C_{\theta, t}(x) = [\theta_{\text{lo}}^\top \widehat{\Phi}(x) - t, \theta_{\text{hi}}^\top \widehat{\Phi}(x)].$$

The specific function class of $\theta$ shown in the thinner lines is a finite set of linear interpolations of the original $\widehat{\theta}_0$ obtained in QR and the new $\widehat{\theta}$ obtained by conformal quantile finetuning, with combination weights within $\{-0.3, -0.2, \ldots, 1.0\}$. The shaded region is then obtained by filling in the area.

## F  RESULTS FOR LABEL PREDICTION SETS ON IMAGENET

Here we present the ImageNet label prediction set experiment abbreviated in Section 5.

**Dataset and model**  We take $K = 9$ large-scale pretrained neural networks on the ImageNet training set (Deng et al., 2009). Our models are {ResNeXt101, ResNet152, ResNet101, DenseNet161, ResNet18, ResNet50, VGG16, Inception, ShuffleNet}, similar as in (Angelopoulos et al., 2020).

We then consider task of constructing label prediction sets with valid coverage and small cardinality. We train and test out conformal procedures on the following two datasets, neither seen by the pretrained models:

(1) ImageNet-Val: The original validation set of ImageNet with 50000 images. We randomly split (varying with seed) this into $|D_{\text{cal}}| = 10000$, $|D_{\text{recal}}| = 10000$, and $|D_{\text{test}}| = 30000$.

(2) ImageNet-V2 (Recht et al., 2019): A new validation set following the roughly the same collection routines of the original images in ImageNet, however believed to have a mild distribution shift and thus slightly harder for classifiers pretrained on ImageNet. This dataset contains 10000 images, which we randomly split (varying with seed) into $|D_{\text{cal}}| = 4000$, $|D_{\text{recal}}| = 1000$, and $|D_{\text{test}}| = 5000$.

**Methods for learning prediction sets**  Our constructions of the prediction sets are based on the Least Ambiguous Set-Valued Classifier (LAC) method of (Sadinle et al., 2019), which turns any base

Table 5: **Results for ImageNet Prediction Sets with Conformal Ensembling.** For each method we report the (test) coverage and set size. Each entry reports the (mean, std) over 8 random seeds.

| | Best conformalized single model | | Conformalized uniform ensemble | | Ensemble via `CP-Gen-Recal` (ours) | |
|---|---|---|---|---|---|---|
| Dataset | Coverage(%) | Size | Coverage(%) | Size | Coverage(%) | Size |
| ImageNet-Val | $90.10 \pm 0.29$ | $1.70 \pm 0.03$ | $90.13 \pm 0.21$ | $1.62 \pm 0.02$ | $90.11 \pm 0.33$ | $\mathbf{1.51 \pm 0.03}$ |
| ImageNetV2 | $90.01 \pm 0.71$ | $5.00 \pm 0.24$ | $89.93 \pm 0.71$ | $4.66 \pm 0.22$ | $90.18 \pm 0.85$ | $\mathbf{4.39 \pm 0.44}$ |

predictor $p$ where $p(\cdot|x)$ denotes the predicted distribution of the $L = 1000$ labels into a prediction set $C_t(x)$ via

$$C_t(x) = \{y \in [L] : p(y|x) > t\},$$

where $t$ is found by conformal prediction.

We consider learning a valid prediction set with smaller set size by finding an optimized *ensemble weight* of the $K$ base predictors using our `CP-Gen-Recal` algorithm. This means we learn prediction sets of the form

$$C_{\theta,t}(x) = \left\{ y \in [L] : p_\theta(y|x) := \sum_{k=1}^{K} \theta_k p_k(y|x) > t \right\},$$

where $\{p_k\}_{k \in [K]}$ are the base predictors.

Our `CP-Gen-Recal` algorithm (and its practical implementation (8)) would solve a primal-dual optimization problem with the efficiency loss and hinge approximate coverage constraint to optimize $(\theta, t)$. However, here the efficiency loss we care about (the cardinality) is non-differentiable. We make a further approximation by considering the $L_q^q$ norm with $q = 0.5$ as the surrogate efficiency loss:

$$\ell_{\text{eff}}(\theta, t; (x_i, y_i)) := \sum_{y'=1}^{L} [p_\theta(y'|x_i) - t]_+^q,$$

with the intuition that the $q \to 0$ limit is exactly the cardinality of $C_{\theta,t}(x_i)$. Our final optimization problem is then

$$\min_{\theta \in \Delta_K, t > 0} \max_{\lambda > 0} \frac{1}{n_{\text{cal}}} \sum_{i=1}^{n_{\text{cal}}} \sum_{y'=1}^{L} [p_\theta(y'|x_i) - t]_+^q + \lambda \frac{1}{n_{\text{cal}}} \sum_{i=1}^{n_{\text{cal}}} \ell_{\text{hinge}}(p_\theta(y_i|x_i) - t).$$

We solve this by SGD on $(\theta, t)$ and (ascent on) $\lambda$, with the softmax parameterization for $\theta$ ($\theta \in \Delta_K$ as an ensemble weight is a probability distribution) and log parametrization for $t > 0$. The learning rate is $10^{-2}$ for $(\theta, t)$ and $10^{-4}$ for $\lambda$. We perform this optimization for 500 epochs over $D_{\text{cal}}$ with batch-size 256 for ImageNet-Val and 64 for ImageNet-V2.

After we obtain the iterates $\left\{ \widehat{\theta}_j \right\}$ (where $j$ denotes the epoch count), we perform a further iterate selection of first re-computing the $\widehat{t}(\widehat{\theta}_j)$ by conformalizing on $D_{\text{cal}}$, and then choosing the iterate $j$ with the best average set size also on $D_{\text{cal}}$, before feeding it into the reconformalization step with $D_{\text{recal}}$. As the $D_{\text{recal}}$ is only used in the reconformalization step, such as method still guarantees valid coverage like the original Algorithm 2.

We compare our above algorithm against two baselines: conformalizing each individual model and reporting the best one, or conformalizing the uniform ensemble (which uses weights $\theta_{\text{unif}} = \frac{1}{K}\mathbf{1}_K$). For these two baselines, for fairness of comparison, we allow them to use the whole $D_{\text{cal}} \cup D_{\text{recal}}$ as the calibration set, as their construction (apart from pre-training) is not data-dependent.

**Results** Table 5 shows that our algorithm is able to learn label prediction sets with valid coverage and improved set sizes over the baselines. This demonstrates the advantage of our method even in applications where the efficiency loss (here set size) is non-differentiable and needs to be further approximated to allow gradient-based algorithms.

# G ABLATION STUDIES

## G.1 CONFORMAL QUANTILE FINETUNING

We report ablation results for the conformal quantile finetuning problem with nominal coverage level $1 - \alpha \in \{80\%, 95\%\}$, and otherwise exactly the same setup as Section 5.1. The conclusions are qualitatively the same as the 90% version presented in Table 1.

Table 6: **Results for conformal quantile finetuning** on real-data regression tasks at level $1 - \alpha = 80\%$. For each method we report the (test) coverage, length, and pinball loss of the corresponding base quantile predictor. All results are averaged over 8 random seeds.

| | CQR | | | QR + CP-Gen-Recal (ours) | | |
|---|---|---|---|---|---|---|
| Dataset | Coverage(%) | Length | $L_{\text{pinball}}^{\text{test}}$ | Coverage(%) | Length | $L_{\text{pinball}}^{\text{test}}$ |
| MEPS_19 | 80.42 | 0.702 | 0.154 | 80.45 | **0.514** | 0.190 |
| MEPS_20 | 80.44 | 0.707 | 0.161 | 80.48 | **0.466** | 0.200 |
| MEPS_21 | 79.91 | 0.696 | 0.151 | 79.85 | **0.618** | 0.192 |
| Facebook_1 | 80.38 | 0.348 | 0.072 | 80.01 | **0.198** | 0.137 |
| Facebook_2 | 79.96 | 0.329 | 0.063 | 79.80 | **0.189** | 0.138 |
| kin8nm | 79.59 | 0.865 | 0.119 | 78.69 | **0.832** | 0.125 |
| naval | 79.91 | 2.777 | 0.311 | 79.76 | **2.721** | 0.311 |
| bio | 80.07 | 1.791 | 0.222 | 80.54 | **1.674** | 0.248 |
| blog_data | 80.64 | 0.399 | 0.082 | 80.10 | **0.272** | 0.158 |
| Nominal $(1 - \alpha)$ | 80.00 | - | - | 80.00 | - | - |

Table 7: **Results for conformal quantile finetuning** on real-data regression tasks at level $1 - \alpha = 95\%$. For each method we report the (test) coverage, length, and pinball loss of the corresponding base quantile predictor. All results are averaged over 8 random seeds.

| | CQR | | | QR + CP-Gen-Recal (ours) | | |
|---|---|---|---|---|---|---|
| Dataset | Coverage(%) | Length | $L_{\text{pinball}}^{\text{test}}$ | Coverage(%) | Length | $L_{\text{pinball}}^{\text{test}}$ |
| MEPS_19 | 94.60 | 1.674 | 0.078 | 95.10 | **1.292** | 0.091 |
| MEPS_20 | 94.72 | 1.650 | 0.081 | 94.78 | **1.261** | 0.097 |
| MEPS_21 | 94.64 | 1.633 | 0.071 | 94.99 | **1.351** | 0.086 |
| Facebook_1 | 94.96 | 0.797 | 0.036 | 95.04 | **0.601** | 0.061 |
| Facebook_2 | 95.17 | 0.700 | 0.031 | 94.98 | **0.560** | 0.060 |
| kin8nm | 95.15 | 1.602 | 0.047 | 94.95 | **1.557** | 0.048 |
| naval | 94.87 | 3.308 | 0.084 | 94.83 | **3.265** | 0.088 |
| bio | 95.17 | 2.698 | 0.073 | 95.22 | **2.587** | 0.084 |
| blog_data | 95.07 | 0.862 | 0.040 | 95.09 | **0.744** | 0.068 |
| Nominal $(1 - \alpha)$ | 95.00 | - | - | 95.00 | - | - |

## G.2 MULTI-OUTPUT REGRESSION

We report ablation results for the multi-output regression problem with nominal coverage level $1 - \alpha \in \{80\%, 95\%\}$, and otherwise exactly the same setup as Section 5.2. The conclusions are qualitatively the same as the 90% version presented in Table 2, except for one dataset at level 95%.

## G.3 COMPARISON OF CP-Gen AND CP-Gen-Recal

We compare the performance of CP-Gen and CP-Gen-Recal on the multi-output regression tasks using the same setup as Section 5.2. Recall that the vanilla CP-Gen optimizes both $(\widehat{\theta}, \widehat{t})$ on $D_{\text{cal}}$ (we additionally reconformalize $\widehat{t}$ on the same $D_{\text{cal}}$ to address the potential bias in $\widehat{t}$ brought by the approximate optimization (8)), whereas our main CP-Gen-Recal algorithm optimizes $\widehat{\theta}$ on $D_{\text{cal}}$ and reconformalizes $\widehat{t}_{\text{recal}}$ on $D_{\text{recal}}$.

Table 10 reports the results. Observe that, except for the volume on one dataset (Humanoid), there is no significant difference in both the coverage and the volume for the two methods. For practice we recommend CP-Gen-Recal whenever the exact coverage guarantee is important, yet this result

Table 8: **Results for multi-output regression** on next-state prediction tasks, at level $1 - \alpha = 80\%$. For each method we report the (test) coverage and volume of its learned box-shaped prediction set. The reported volume is the "halfened" version $\prod_{i=1}^{d_{\text{out}}} u_i$. All results are averaged over 8 random seeds.

| | Coord-wise | | Coord-wise-Recal | | CP-Gen-Recal (ours) | |
| Dataset | Coverage(%) | Volume | Coverage(%) | Volume | Coverage(%) | Volume |
|---|---|---|---|---|---|---|
| Cartpole | 87.82 | $1.74 \times 10^{-6}$ | 80.08 | $7.83 \times 10^{-7}$ | 80.09 | $\mathbf{7.45 \times 10^{-7}}$ |
| Half-Cheetah | 88.28 | $4.26 \times 10^{-7}$ | 79.96 | $2.42 \times 10^{-8}$ | 80.04 | $\mathbf{1.37 \times 10^{-8}}$ |
| Ant | 87.77 | $1.97 \times 10^{-5}$ | 80.12 | $3.97 \times 10^{-7}$ | 80.06 | $\mathbf{1.75 \times 10^{-7}}$ |
| Walker | 90.25 | $5.88 \times 10^{-7}$ | 80.28 | $3.13 \times 10^{-9}$ | 80.28 | $\mathbf{1.45 \times 10^{-9}}$ |
| Swimmer | 91.49 | $1.33 \times 10^{-6}$ | 79.99 | $6.18 \times 10^{-8}$ | 79.97 | $\mathbf{4.32 \times 10^{-9}}$ |
| Hopper | 86.47 | $3.25 \times 10^{-10}$ | 79.87 | $7.40 \times 10^{-11}$ | 79.97 | $\mathbf{4.41 \times 10^{-11}}$ |
| Humanoid | 90.84 | $2.86 \times 10^{-7}$ | 80.05 | $9.47 \times 10^{-13}$ | 80.02 | $\mathbf{2.41 \times 10^{-13}}$ |
| Nominal $(1 - \alpha)$ | 80.00 | - | 80.00 | - | 80.00 | - |

Table 9: **Results for multi-output regression** on next-state prediction tasks, at level $1 - \alpha = 95\%$. For each method we report the (test) coverage and volume of its learned box-shaped prediction set. The reported volume is the "halfened" version $\prod_{i=1}^{d_{\text{out}}} u_i$. All results are averaged over 8 random seeds.

| | Coord-wise | | Coord-wise-Recal | | CP-Gen-Recal (ours) | |
| Dataset | Coverage(%) | Volume | Coverage(%) | Volume | Coverage(%) | Volume |
|---|---|---|---|---|---|---|
| Cartpole | 97.21 | $1.07 \times 10^{-4}$ | 95.10 | $4.60 \times 10^{-5}$ | 95.12 | $\mathbf{8.61 \times 10^{-6}}$ |
| Half-Cheetah | 96.80 | $2.37 \times 10^{-4}$ | 95.03 | $4.03 \times 10^{-5}$ | 95.01 | $\mathbf{3.29 \times 10^{-5}}$ |
| Ant | 96.65 | $5.30 \times 10^{-1}$ | 95.02 | $4.87 \times 10^{-2}$ | 95.09 | $\mathbf{2.39 \times 10^{-2}}$ |
| Walker | 97.01 | $8.08 \times 10^{-4}$ | 94.94 | $6.21 \times 10^{-5}$ | 94.99 | $\mathbf{4.27 \times 10^{-5}}$ |
| Swimmer | 97.74 | $3.44 \times 10^{-4}$ | 94.95 | $3.77 \times 10^{-5}$ | 95.01 | $\mathbf{5.34 \times 10^{-6}}$ |
| Hopper | 96.27 | $1.76 \times 10^{-8}$ | 94.96 | $\mathbf{8.23 \times 10^{-9}}$ | 94.96 | $1.19 \times 10^{-8}$ |
| Humanoid | 97.22 | $3.58 \times 10^{-1}$ | 94.99 | $7.69 \times 10^{-4}$ | 94.91 | $\mathbf{7.49 \times 10^{-4}}$ |
| Nominal $(1 - \alpha)$ | 95.00 | - | 95.00 | - | 95.00 | - |

shows that—perhaps originating from the fact that here $n_{\text{cal}} = 20000$ is large—the coverage (generalization error) of `CP-Gen` is also nearly valid, which may be better than what our Proposition 2 suggests.

Table 10: Comparison of `CP-Gen` and `CP-Gen-Recal` on the multi-output regression tasks. The reported volume is the "halfened" version $\prod_{i=1}^{d_{\text{out}}} u_i$.

| | CP-Gen-Recal | | CP-Gen | |
| Dataset | Coverage(%) | Volume | Coverage(%) | Volume |
|---|---|---|---|---|
| Cartpole | 90.12 | $2.30 \times 10^{-6}$ | 90.09 | $2.30 \times 10^{-6}$ |
| Half-Cheetah | 90.02 | $9.07 \times 10^{-7}$ | 89.96 | $8.83 \times 10^{-7}$ |
| Ant | 90.02 | $8.25 \times 10^{-5}$ | 89.98 | $8.21 \times 10^{-5}$ |
| Walker | 89.94 | $3.47 \times 10^{-7}$ | 89.91 | $3.30 \times 10^{-7}$ |
| Swimmer | 90.13 | $1.46 \times 10^{-7}$ | 89.96 | $1.29 \times 10^{-7}$ |
| Hopper | 89.92 | $8.25 \times 10^{-10}$ | 89.92 | $8.23 \times 10^{-10}$ |
| Humanoid | 89.94 | $4.95 \times 10^{-8}$ | 90.05 | $7.08 \times 10^{-8}$ |
| Nominal | 90.00 | - | 90.00 | - |

# H ADDITIONAL EXPERIMENTS AND ANALYSES

## H.1 CONDITIONAL COVERAGE OF `CP-Gen-Recal`

We analyze the improved length prediction intervals learned by `CP-Gen-Recal` (Section 5.1) by evaluating its *conditional* coverage metrics and comparing with the baseline `CQR` method.

As conditional coverage is hard to reliably estimate from finite data, we consider two proxy metrics proposed in (Feldman et al., 2021) that measure the independence between *length* and *indicator of coverage*:

- The correlation coefficient (Corr) between the following two random variables: the interval size $L = \text{length}(C(X))$ and the indicator of coverage $V = \mathbf{1}\left\{Y \in \widehat{C}(X)\right\}$. A (population) correlation of 0 is a necessary (but not sufficient) condition of perfect conditional coverage (Feldman et al., 2021). Here we measure the absolute correlation, which is smaller the better.
- HSIC: A more sophisticated correlation metric between $L$ and $V$ that takes into account nonlinear correlation structures. A (population) HSIC of 0 is a necessary and sufficient condition of the independence between $L$ and $V$. We estimate HSIC on the finite test data using the method in (Feldman et al., 2021).

Table 11 reports the results. Observe that while our `CP-Gen-Recal` improves the length, it achieves worse (higher) Correlation/HSIC than the baseline `CQR`, which is expected as length and conditional coverage often come as a trade-off.

Table 11: **Conditional coverage results for conformal quantile finetuning** on real-data regression tasks at level $1 - \alpha = 90\%$. For each method we report the (absolute) correlation coefficient as well as the HSIC metric between length and indicator of coverage. All results are averaged over 8 random seeds.

| Dataset | CQR | | | QR + CP-Gen-Recal (ours) | | |
|---|---|---|---|---|---|---|
| | Corr($\downarrow$) | HSIC($\downarrow$) | Length($\downarrow$) | Corr($\downarrow$) | HSIC($\downarrow$) | Length($\downarrow$) |
| MEPS_19 | **0.022** | **3.03 × 10⁻⁵** | 1.167 | 0.049 | $1.77 \times 10^{-4}$ | **0.890** |
| MEPS_20 | **0.032** | **3.63 × 10⁻⁵** | 1.165 | 0.113 | $2.66 \times 10^{-4}$ | **0.830** |
| MEPS_21 | **0.029** | **4.72 × 10⁻⁵** | 1.145 | 0.068 | $2.20 \times 10^{-4}$ | **0.962** |
| Facebook_1 | **0.029** | **1.27 × 10⁻⁵** | 0.555 | 0.175 | $7.34 \times 10^{-4}$ | **0.384** |
| Facebook_2 | **0.024** | **1.16 × 10⁻⁵** | 0.491 | 0.116 | $2.68 \times 10^{-4}$ | **0.364** |
| kin8nm | **0.031** | **4.85 × 10⁻⁵** | 1.214 | 0.084 | $9.32 \times 10^{-5}$ | **1.173** |
| naval | 0.091 | **1.05 × 10⁻⁵** | 3.095 | **0.064** | $2.16 \times 10^{-5}$ | **3.077** |
| bio | **0.026** | **4.15 × 10⁻⁵** | 2.271 | 0.041 | $1.09 \times 10^{-4}$ | **2.164** |
| blog_data | **0.013** | **4.60 × 10⁻⁵** | 0.605 | 0.141 | $5.75 \times 10^{-4}$ | **0.496** |

## H.2 ALTERNATIVE TWEAKS FOR `CQR` BASELINE

Here we test two additional tweaked versions of the `CQR` baseline in the prediction interval experiment of Section 5.1:

- `CQR`-$D_{\text{train}} \cup D_{\text{cal}}$: Use dataset $D_{\text{train}} \cup D_{\text{cal}}$ for training the base quantile regressor, then conformalize on $D_{\text{recal}}$.
- `CQR-PinballFt`: Train the base quantile regressor on $D_{\text{train}}$, and finetune the last linear layer on $D_{\text{cal}}$ using the pinball loss (same as training), and conformalize on $D_{\text{recal}}$.

Optimization details about these two methods are described in Section H.2.1.

**Result** Table 12 reports the results for these two tweaked baselines, in comparison with our original baseline `CQR` as well as our proposed `QR + CP-Gen-Recal`. Observe that using more training data (`CQR`-$D_{\text{train}} \cup D_{\text{cal}}$) improves the length slightly on some datasets but not all. In contrast, `CQR-PinballFt` is unable to improve either the pinball loss or the length over the base `CQR`(observe that `CQR-PinballFt` uses the same set of training data as `CQR`-$D_{\text{train}} \cup D_{\text{cal}}$ but uses a less expressive model in the finetuning stage). Overall, on almost all datasets (except for kin8nm), our `CP-Gen-Recal` still achieves the best length.

Table 12: **Results for conformal quantile finetuning** on real-data regression tasks at level $1-\alpha = 90\%$. Here we compare our `CP-Gen-Recal` method with **tweaked versions of the baseline** `CQR` **method**. All results are averaged over the same 8 random seeds as in Table 1. All (average) coverages are within $(90 \pm 0.5)\%$ and omitted here.

| Dataset | CQR-$D_{\text{train}}$ | | CQR-$D_{\text{train}} \cup D_{\text{cal}}$ | | CQR-PinballFt | | QR + CP-Gen-Recal (ours) | |
|---|---|---|---|---|---|---|---|---|
| | Length | $L_{\text{pinball}}^{\text{test}}$ | Length | $L_{\text{pinball}}^{\text{test}}$ | Length | $L_{\text{pinball}}^{\text{test}}$ | Length | $L_{\text{pinball}}^{\text{test}}$ |
| MEPS_19 | 1.167 | 0.112 | 1.171 | 0.111 | 1.192 | 0.112 | **0.890** | 0.131 |
| MEPS_20 | 1.165 | 0.117 | 1.179 | 0.114 | 1.190 | 0.117 | **0.830** | 0.141 |
| MEPS_21 | 1.145 | 0.107 | 1.150 | 0.106 | 1.249 | 0.107 | **0.962** | 0.129 |
| Facebook_1 | 0.555 | 0.052 | 0.549 | 0.051 | 0.578 | 0.052 | **0.384** | 0.090 |
| Facebook_2 | 0.491 | 0.044 | 0.472 | 0.042 | 0.523 | 0.044 | **0.364** | 0.092 |
| kin8nm | 1.214 | 0.076 | **1.165** | 0.072 | 1.232 | 0.075 | 1.173 | 0.078 |
| naval | 3.095 | 0.164 | 3.089 | 0.164 | 3.096 | 0.164 | **3.077** | 0.166 |
| bio | 2.271 | 0.130 | 2.240 | 0.128 | 2.271 | 0.130 | **2.164** | 0.148 |
| blog_data | 0.605 | 0.058 | 0.551 | 0.056 | 0.660 | 0.058 | **0.496** | 0.107 |

### H.2.1 OPTIMIZATION DETAILS

`CQR`-$D_{\text{train}} \cup D_{\text{cal}}$: Our original `CQR` baseline used $D_{\text{cal}}$ for monitoring validation loss and automatically determining the early stopping (cf. Section E.1), and $D_{\text{recal}}$ for conformalization. To optimize on $D_{\text{train}} \cup D_{\text{cal}}$, we do not use automatic learning rate decay and early stopping, but instead manually picked the number of epochs and corresponding learning rate decay schedule that is close to average runs of the `CQR` method on each dataset. This choice ensures that our new baseline still gets to use see the exact same amount of data for conformalizing ($D_{\text{recal}}$) and testing ($D_{\text{test}}$), and has a optimization setup as close as possible the original `CQR` baseline.

More concretely, we optimize for 800 epochs for {MEPS_19, MEPS_20, MEPS_21}, 1500 epochs for {Facebook_1, Facebook_2, blog_data}, 6000 epochs for kin8nm, 350 epochs for naval, and 2500 epochs for bio. For all datasets, the learning rate decays by 10x twice, at 90% and 95% of the total epochs.

`CQR-PinballFt`: We finetuned on $D_{\text{cal}}$ with 1000 epochs and batch size 256. The learning rate was chosen within $\{10^{-2}, 10^{-3}\}$ and the results are not too different for these two choices (length difference is within 0.010 for these two choices, and there is no overall winner). We presented the results with learning rate $10^{-3}$.

### H.3 ADDITIONAL `Max-score-Conformal` BASELINE FOR MULTI-OUTPUT REGRESSION

We test one additional baseline method for the multi-output regression experiment in Section 5.2:

- `Max-score-Conformal`: Here we consider the *hypercube*-shaped predictor

$$C_t(x) = \prod_{i=1}^{d_{\text{out}}} [\widehat{f}_i(x) - t, f_i(x) + t], \tag{16}$$

  and use conformal prediction on $D_{\text{cal}} \cup D_{\text{recal}}$ to compute a conformalized $\widehat{t}$ and the final prediction set $C_{\widehat{t}}(x)$. In other words, we perform standard conformal prediction with score function $\|y - \widehat{f}(x)\|_\infty$.

We remark that both the `Coord-wise-Recal` and the `Max-score-Conformal` baseline methods are special instances of `CP-Gen-Recal` with some fixed $\theta_0$: In our parametrization, $u = (\theta, t)$, $\theta$ determines the *shape* (i.e. relative ratios between the $u_i's$) whereas $t$ determines the *size*. Therefore, `Max-score-Conformal` can be thought of as choosing $\theta_0$ to be the all-ones ratio (i.e. hypercube-shaped), whereas `Coord-wise-Recal` can be thought of as choosing $\theta_0$ from a coordinate-wise one dimensional conformal prediction.

**Result** Table 13 reports the result for `Max-score-Conformal`. Compared with the existing baseline `Coord-wise-Recal`, `Max-score-Conformal` achieves better volume on the Cartpole dataset but worse volume on almost all other datasets (except for Swimmer where their

volumes are similar). Further, note that `Max-score-Conformal` achieves significantly higher volumes for certain datsets (Ant, Humanoid). Our inspection shows that this due to the fact that there are a certain number of hard-to-predict state dimensions (and many other easy-to-predict state dimensions) for these two datasets. Therefore, `Coord-wise-Recal` which builds on `Coord-wise` adapts to this structure and uses only a high length on these dimensions only, whereas the `Max-score-Conformal` method pays this max conformal score on all dimensions to yield an unnecessarily high volume.

We remark that our `CP-Gen-Recal` still performs significantly better than both baselines.

Table 13: **Results for multi-output regression** on next-state prediction tasks, at level $1 - \alpha = 90\%$. The setting is the same as in Table 2 (with the same 8 random seeds), and here we compare additionally with the `Max-score-Conformal` baseline method described in (16).

| Dataset | Coord-wise-Recal | | Max-score-Conformal | | CP-Gen-Recal (ours) | |
|---|---|---|---|---|---|---|
| | Coverage(%) | Volume | Coverage(%) | Volume | Coverage(%) | Volume |
| Cartpole | 90.17 | $5.10 \times 10^{-6}$ | 90.10 | $3.07 \times 10^{-6}$ | 90.12 | $\mathbf{2.30 \times 10^{-6}}$ |
| Half-Cheetah | 90.06 | $1.23 \times 10^{-6}$ | 89.96 | $1.72 \times 10^{-4}$ | 90.02 | $\mathbf{9.07 \times 10^{-7}}$ |
| Ant | 89.99 | $1.70 \times 10^{-4}$ | 90.06 | $3.46 \times 10^{2}$ | 90.02 | $\mathbf{8.25 \times 10^{-5}}$ |
| Walker | 90.01 | $7.33 \times 10^{-7}$ | 90.02 | $1.03 \times 10^{-2}$ | 89.94 | $\mathbf{3.47 \times 10^{-7}}$ |
| Swimmer | 89.90 | $2.22 \times 10^{-6}$ | 90.08 | $2.21 \times 10^{-6}$ | 90.13 | $\mathbf{1.46 \times 10^{-7}}$ |
| Hopper | 90.02 | $1.01 \times 10^{-9}$ | 89.96 | $1.29 \times 10^{-8}$ | 89.92 | $\mathbf{8.25 \times 10^{-10}}$ |
| Humanoid | 89.95 | $8.53 \times 10^{-8}$ | 89.98 | $2.48 \times 10^{7}$ | 89.94 | $\mathbf{4.95 \times 10^{-8}}$ |
| Nominal $(1-\alpha)$ | 90.00 | - | 90.00 | - | 90.00 | - |

### H.4 100 RANDOM SEEDS AND STANDARD DEVIATION

Here we repeat the experiments in Section 5.1 & 5.2 with 100 random seeds and the exact same setups. We report the mean and standard deviations in Table 14 for the prediction intervals experiment (Section 5.1), Table 15 for the mean and Table 16 for the standard deviation for the multi-output regression experiment (Section 5.2).

Table 14: **Results for conformal quantile finetuning** on real-data regression tasks at level $1-\alpha = 90\%$. For each method we report the (test) coverage, length, and pinball loss of the corresponding base quantile predictor. All results are averaged over 100 random seeds.

| Dataset | CQR | | | QR + CP-Gen-Recal (ours) | | |
|---|---|---|---|---|---|---|
| | Coverage(%) | Length | $L^{\text{test}}_{\text{pinball}}$ | Coverage(%) | Length | $L^{\text{test}}_{\text{pinball}}$ |
| MEPS_19 | 89.92 ±1.16 | 1.147 ±0.057 | 0.107 ±0.013 | 89.95 ±0.012 | $\mathbf{0.895 \pm 0.126}$ | 0.130 ±0.015 |
| MEPS_20 | 89.90 ±0.98 | 1.164 ±0.054 | 0.109 ±0.012 | 89.97 ±0.011 | $\mathbf{0.872 \pm 0.113}$ | 0.131 ±0.015 |
| MEPS_21 | 90.00 ±0.94 | 1.162 ±0.056 | 0.104 ±0.011 | 90.08 ±0.011 | $\mathbf{0.910 \pm 0.133}$ | 0.126 ±0.013 |
| Facebook_1 | 90.12 ±0.71 | 0.540 ±0.040 | 0.050 ±0.007 | 90.07 ±0.007 | $\mathbf{0.382 \pm 0.051}$ | 0.089 ±0.009 |
| Facebook_2 | 90.04 ±0.50 | 0.497 ±0.028 | 0.044 ±0.005 | 90.06 ±0.005 | $\mathbf{0.389 \pm 0.075}$ | 0.091 ±0.007 |
| kin8nm | 90.34 ±1.37 | 1.238 ±0.067 | 0.076 ±0.004 | 90.31 ±0.013 | $\mathbf{1.216 \pm 0.068}$ | 0.080 ±0.004 |
| naval | 89.99 ±1.13 | 3.101 ±0.015 | 0.164 ±0.001 | 89.95 ±0.011 | $\mathbf{3.095 \pm 0.028}$ | 0.167 ±0.001 |
| bio | 90.00 ±0.69 | 2.261 ±0.033 | 0.130 ±0.002 | 89.97 ±0.005 | $\mathbf{2.154 \pm 0.031}$ | 0.148 ±0.003 |
| blog_data | 89.99 ±0.60 | 0.593 ±0.033 | 0.058 ±0.005 | 89.95 ±0.007 | $\mathbf{0.460 \pm 0.075}$ | 0.104 ±0.006 |
| Nominal $(1-\alpha)$ | 90.00 | - | - | 90.00 | - | - |

Table 15: **Results for multi-output regression (mean)** on next-state prediction tasks, at level $1 - \alpha = 90\%$. For each method we report the (test) coverage and volume of its learned box-shaped prediction set. The reported volume is the "halfened" version $\prod_{i=1}^{d_{\text{out}}} u_i$. All results are averaged over 100 random seeds.

| | Coord-wise | | Coord-wise-Recal | | CP-Gen-Recal (ours) | |
|---|---|---|---|---|---|---|
| Dataset | Coverage(%) | Volume | Coverage(%) | Volume | Coverage(%) | Volume |
| Cartpole | 94.30 | $1.14 \times 10^{-5}$ | 90.02 | $4.80 \times 10^{-6}$ | 90.03 | $\mathbf{2.05 \times 10^{-6}}$ |
| Half-Cheetah | 93.84 | $1.05 \times 10^{-5}$ | 90.00 | $1.22 \times 10^{-6}$ | 90.02 | $\mathbf{9.01 \times 10^{-7}}$ |
| Ant | 93.53 | $3.26 \times 10^{-3}$ | 89.94 | $1.75 \times 10^{-4}$ | 89.98 | $\mathbf{9.22 \times 10^{-5}}$ |
| Walker | 94.52 | $2.82 \times 10^{-5}$ | 89.99 | $7.48 \times 10^{-7}$ | 90.00 | $\mathbf{3.74 \times 10^{-7}}$ |
| Swimmer | 95.65 | $2.82 \times 10^{-5}$ | 90.02 | $2.18 \times 10^{-6}$ | 90.01 | $\mathbf{1.24 \times 10^{-7}}$ |
| Hopper | 92.95 | $2.53 \times 10^{-9}$ | 89.98 | $8.86 \times 10^{-10}$ | 89.99 | $\mathbf{7.04 \times 10^{-10}}$ |
| Humanoid | 94.87 | $7.43 \times 10^{-4}$ | 90.06 | $1.69 \times 10^{-7}$ | 90.03 | $\mathbf{8.95 \times 10^{-8}}$ |
| Nominal $(1-\alpha)$ | 90.00 | - | 90.00 | - | 90.00 | - |

Table 16: **Results for multi-output regression (standard deviation)** on next-state prediction tasks, at level $1 - \alpha = 90\%$. For each method we report the (test) coverage and volume of its learned box-shaped prediction set. The reported volume is the "halfened" version $\prod_{i=1}^{d_{\text{out}}} u_i$. All standard deviations are computed over 100 random seeds.

| | Coord-wise | | Coord-wise-Recal | | CP-Gen-Recal (ours) | |
|---|---|---|---|---|---|---|
| Dataset | Coverage(%) | Volume | Coverage(%) | Volume | Coverage(%) | Volume |
| Cartpole | 0.35 | $3.57 \times 10^{-6}$ | 0.31 | $1.33 \times 10^{-6}$ | 0.27 | $4.46 \times 10^{-7}$ |
| Half-Cheetah | 0.25 | $2.46 \times 10^{-6}$ | 0.32 | $2.75 \times 10^{-7}$ | 0.31 | $2.10 \times 10^{-7}$ |
| Ant | 0.27 | $1.50 \times 10^{-3}$ | 0.29 | $8.06 \times 10^{-5}$ | 0.29 | $4.15 \times 10^{-5}$ |
| Walker | 0.24 | $8.81 \times 10^{-6}$ | 0.30 | $2.18 \times 10^{-7}$ | 0.29 | $1.12 \times 10^{-7}$ |
| Swimmer | 0.24 | $6.47 \times 10^{-6}$ | 0.29 | $5.83 \times 10^{-7}$ | 0.33 | $3.36 \times 10^{-8}$ |
| Hopper | 0.29 | $6.83 \times 10^{-10}$ | 0.29 | $2.26 \times 10^{-10}$ | 0.33 | $1.97 \times 10^{-10}$ |
| Humanoid | 0.23 | $1.22 \times 10^{-3}$ | 0.30 | $2.71 \times 10^{-7}$ | 0.29 | $1.47 \times 10^{-7}$ |

