# OpenReview forum: "Efficient and Differentiable Conformal Prediction with General Function Classes"
_ICLR.cc/2022/Conference — ICLR 2022 Poster_

### Official Review · Reviewer_3LHz · 2021-10-27

**Correctness:** 3
**Technical Novelty And Significance:** 4
**Empirical Novelty And Significance:** 3
**Recommendation:** 8
**Confidence:** 4

**Main Review:**

In my review I would like to address each of the evaluation criteria.

Technical novelty and significance:

The presented idea is novel and interesting. It is well supported by theoretical arguments and experimental results. It is interesting to see that the bound on coverage becomes weaker when the complexity of the space over which is optimized increases. This is in fact a natural result.

The presented idea is in fact quite similar to this ICML paper: https://arxiv.org/pdf/1802.07167.pdf. Similarly as the Lagrange formulation in Eq. 8, in that paper an optimization problem that incorporates both efficiency and coverage is introduced, but the objective function is a bit different. It would be interesting to compare empirically with that method.

Empirical novelty and significance:

The experiments show what they need to show, i.e. that the proposed method is better than conformalized quantile regression. However, the results reported for CQR are a bit worse than those reported in https://arxiv.org/pdf/2107.00363.pdf (look for the datasets that are reported in the two papers). It is not so clear why that is the case.

The experiments for multi-output regression and multi-class classification are a nice add-on to illustrate the broad applicability of the presented method, but they are not essential for the paper.

Correctness:

The theoretical look correct, but at some stages the readability could be improved:

Proposition 2a looks a bit awkward because the inequality has on the left side an average over all randomly drawn calibration sets, whereas the right hand side is about a specific calibration set. I would suggest to add "for any calibration set D_cal" to the beginning of the proposition.
I don't understand why epsilon_zero is included in Algorithm 1. The authors mention in Section 3.1 "for analysis purposes", but which analysis is this? Perhaps this could be removed to improve readability.
The experimental results have some minor issues w.r.t. which baselines are considered and how these baselines are tuned (see above). However, I don't see a problem with the main message of the paper.

**Summary Of The Paper:**

This paper introduces an extension of conformal prediction with a different formulation. Instead of guaranteed coverage for a finite calibration set, the authors solve a constrained optimization problem where the length of prediction intervals is minimized subject to a coverage constraint. As a result, coverage is only asymptotically guaranteed, but instead the length of the prediction intervals can be shortened compared to traditional split conformal prediction.

The authors present theoretical results in the form of coverage bounds for specific function classes.

In the experiments classical regression datasets are analyzed, as well as multi-output regression problems and one multi-class classification problem.

**Summary Of The Review:**

Interesting paper. Some minor issues w.r.t. comparison with existing work.

---

> ### Author Response · Authors · 2021-11-22
> **Response to Reviewer 3LHz**
>
> Thank you for your positive feedback on our paper! We have performed several additional experiments and analyses, and revised our paper to incorporate various reviewer suggestions. All changes except for fixed typos are marked in red for clarity.
>
> We respond to your comments as follows.
>
> > The presented idea is in fact quite similar to this ICML paper: https://arxiv.org/pdf/1802.07167.pdf… It would be interesting to compare empirically with that method.
>
> Thank you for pointing out this related work. We think their approach is indeed similar to ours in their risk minimization method for finding a short prediction interval with good coverage; however, their method does not use conformal prediction and hence does not guarantee $1-\alpha$ finite-sample coverage. Also, their experiments only minimized the length and did not consider other efficiency losses.
>
> We have properly cited this paper in our revision, and we believe a further comparison between their method and conformal methods (such as CQR and ours) is indeed a good direction for future work.
>
>
> > The results reported for CQR are a bit worse than those reported in https://arxiv.org/pdf/2107.00363.pdf (look for the datasets that are reported in the two papers). It is not so clear why that is the case.
>
> We compared this very recent paper and ours, and we noticed various differences between the architectures (one-hidden-layer vs. two-hidden-layer networks) and optimization methods (Adam vs. SGD), all of which may have caused the difference in terms of the lengths. Although their paper provided some implementation details, we are not yet able to find their open source implementation. We believe a more careful reimplementation of their work and a thorough comparison may be a good thing to try out for future studies.
>
> We remark that our data standardization and CQR optimization procedure is more similar to Feldman et al. (2021) (our setup partly follows theirs), and our reported lengths for CQR are indeed similar with theirs on the MEPS datasets and kin8nm (for the other datasets we used slightly different pre-processing routines).
>
> > Proposition 2a looks a bit awkward because the inequality has on the left side an average over all randomly drawn calibration sets, whereas the right hand side is about a specific calibration set. I would suggest to add "for any calibration set D_cal" to the beginning of the proposition.
>
> In our original version, the definition of the population losses $L_{\rm coverage}$ and $L_{\rm eff}$ may have been a bit confusing (this was also pointed out by reviewer bDLk). We have revised the definitions before Proposition 2 accordingly. Proposition 2a now has the population coverage loss on the left hand side, and uniform convergence of coverage on the right hand side, similar to the usual conversion from uniform convergence to generalization bounds in the standard analysis of empirical risk minimization.
>
> > The authors mention in Section 3.1 "for analysis purposes", but which analysis is this? Perhaps this could be removed to improve readability.
>
> By the “analysis” we meant the theoretical results in Proposition 2b and Proposition 7b, which require $\varepsilon_0$ to be higher than the uniform convergence quantity $\varepsilon_{\rm coverage}$, in order to achieve near-optimal efficiency losses. We have clarified this in our revision.

---

### Official Review · Reviewer_bDLk · 2021-11-01

**Correctness:** 4
**Technical Novelty And Significance:** 3
**Empirical Novelty And Significance:** 3
**Recommendation:** 6
**Confidence:** 4

**Main Review:**

There are a number of strengths that I quite like about this paper:
- The paper is exceptionally well written and clear to read.
- The core idea itself is appealing: strict adherence to proving marginal (or other) coverage for set-based classifiers, which generally relies on preserving exchangeability, is generally a good thing, but can be limiting/overly conservative. It is useful to formalize what practitioners stand to gain/lose by directly optimizing conformal predictors for efficiency.
- The framework itself is fairly simple (which is a nice thing), and empirically improves over a good CQR baseline.

There are, however, some weaknesses, and I do have a few questions/concerns.

- A minor point: though it is nice to formalize Propositions 2-4 (and they are nicely stated and proven), from a practical point of view $\epsilon_{\mathrm{coverage}}$ and $\epsilon_{\mathrm{eff}}$ are still quite loosely bounded and its unclear how much value they really add. As a practitioner, as a rule of thumb I already know that my train performance will roughly generalize well to my test performance if my model is low capacity and my data sample is large. The bounds given in Props. 3 & 4 might be too large to be practically useful, i.e., if I was setting $\alpha' = \alpha - \epsilon_{\mathrm{coverage}}$ based on this analysis. (I also appreciate the use of absolute constant C to clean up notation, but it does make the bound looser/not directly computable).
That said, I understand that as positioned in this paper, these theoretical results are intended to formalize that stated "rule of thumb", i.e, that "we may expect $\texttt{CP-Gen}$ to generalize well [...] whenever it learns $K \ll n_{\mathrm{cal}}$ parameters," though it may be good to qualify this result more.

- I don't think this was mentioned: the proof of Proposition 2 (and others) relies on $D_\mathrm{cal}$ being i.i.d., which is slightly stricter than exchangeability (required for standard CP).

- It is interesting from Table 1 that the quantile loss increases after optimizing $\texttt{CP-Gen}$. An advantage of CQR is that it is adaptive to local variability. In this sense, it can achieve better approximate conditional coverage than a fixed interval. Do you lose any of this when you train $\texttt{CP-Gen}$?

- The length numbers in Table 1 seem quite a bit lower than those reported in the CQR paper (for example, Fig. 3 in CQR lists the best meps_19 result at 2.36, vs the 1.167 reported in your Table 1). Can you explain this difference? At the same time, it would be good to understand how much optimizing for $\texttt{CP-Gen}$ actually helps over choosing better base predictors with favorable properties (e.g., see [Conformal Prediction using Conditional Histograms](https://arxiv.org/abs/2105.08747) which improves over CQR on many of the same datasets reported here).



=== Minor Comments ===

- At the start of Sec. 3.2 you define $L_{\mathrm{eff}, \mathrm{coverage}}$  as the expected i.i.d. sample mean, which is equal to the true population average. However, I do find the notation slightly confusing (in terms of expectation over $D_{\mathrm{cal}}$) with what would be the train accuracy on $D_{\mathrm{cal}}$ after solving the optimization problem. In Section B.2 you write L in terms of $\mathbb{E}[l_\mathrm{eff} (C_{\theta, t}); (X, Y))]$ which I find preferable and would recommend that you could simply use here as well.

-  Typo: when bounding $\epsilon_{\mathrm{eff}}$ in the proof Prop. 3, it should be $L_{\mathrm{eff}}$ instead of $L_{\mathrm{coverage}}$.

- For clarity, it might be good to formally define $\epsilon_i$ as Rademacher r.v.'s when applying the symmetrization argument in the proof of Prop 3.

- I would suggest reconsidering the use of "exact coverage" in Section 3.3, as this can be conflated for the case where the bound $\mathbb{P}(Y \in C(X)) \geq 1 - \alpha$ holds with equality.

- Typos, bottom of page 7: "$\hat{t}_{\mathrm{recal}}$ to _guarantee_ coverage" and "emphasize that the _approximation_ in (8)".

- Typo, Section 5.1: I believe you mean to include $+ t$ in the upper interval bound of $C_{\theta, t}$.

**Summary Of The Paper:**

This paper generalizes the standard conformal prediction calibration setup to a constrained empirical risk minimization problem. Specifically, this work seeks to optimize some efficiency loss, while satisfying coverage constraints. This formulation allows for the introduction of multiple, learnable parameters which can help find a better set-based predictor. The paper explains the implications of this approach by analyzing the generalization error that may occur when transferring the solution learned by constrained ERM to a test population. It also explains practical ways of learning this problem via differentiable surrogate losses and Lagrangians. Contributions-wise, the paper contributes validating theoretical analysis that proves that this method can achieve approximate coverage and near optimal efficiency for certain set-function classes. It also empirically shows that the proposed method can improve over baselines that are not directly optimized for efficiency.

**Summary Of The Review:**

In general, the paper is well-written and the idea is appealing. As mentioned in the main review, it would be good to understand more about the solutions that the proposed method finds and what (if anything) they might be sacrificing (e.g., conditional coverage). It would also be good to compare to somewhat more recent baselines (e.g., CHR for regression, RAPS for classification).

---

> ### Author Response · Authors · 2021-11-22
> **Response to Reviewer Reviewer bDLk**
>
> Thank you for your positive feedback and the thoughtful comments on our paper! We added some additional experiments and have revised our paper according to your various suggestions; all modifications other than fixed typos are marked in red for clarity.
>
> We respond to your comments as follows.
>
> > I don't think this was mentioned: the proof of Proposition 2 (and others) relies on Dcal being i.i.d., which is slightly stricter than exchangeability (required for standard CP).
>
> We agree the i.i.d.-ness is required in the proof, and it is good to point this out explicitly. We have added a sentence mentioning this before Proposition 2.
>
> > It is interesting from Table 1 that the quantile loss increases after optimizing CP-Gen. An advantage of CQR is that it is adaptive to local variability. In this sense, it can achieve better approximate conditional coverage than a fixed interval. Do you lose any of this when you train CP-Gen?
>
> Thank you for this insightful question; we agree some analyses e.g. on the conditional coverage could be helpful for understanding the behavior of our CP-Gen algorithm.
>
> In Appendix H.1, we evaluated some conditional coverage metrics for both CQR and our CP-Gen on the prediction interval experiments in Section 5.1. We find that CP-Gen---while improving the length over CQR---indeed achieves slightly worse conditional coverage. This is as expected, as conditional coverage is a more stringent requirement than marginal coverage, to achieve a better conditional coverage one typically needs to pay in terms of the length (this was reported in e.g. Table 2 & 3 of Feldman et al. (2021)), and the reverse is probably also true.
>
> > The length numbers in Table 1 seem quite a bit lower than those reported in the CQR paper (for example, Fig. 3 in CQR lists the best meps_19 result at 2.36, vs the 1.167 reported in your Table 1). Can you explain this difference?
>
> We compared our implementation with the CQR paper. Although our architectures are the same, we found various differences in terms of both the optimization procedure (difference batch sizes, Adam vs. SGD optimizer), as well as the standardization (they standardized the output by the mean absolute value whereas we used the standard deviation). Therefore the lengths between ours and the CQR paper are incomparable in general.
>
> We remark that our data standardization and CQR optimization procedure is more similar to Feldman et al. (2021) (our setup partly follows theirs), and our reported lengths for CQR are indeed similar with theirs on the MEPS datasets and kin8nm (for the other datasets we used slightly different pre-processing routines).
>
> > At the start of Sec. 3.2 you define $L_{{\rm eff}, {\rm coverage}}$ as the expected i.i.d. sample mean… I do find the notation slightly confusing… In Section B.2 you write L in terms of $E[\ell_{\rm eff}(C_{\theta, t}); (X,Y))]$ which I find preferable and would recommend that you could simply use here as well.
>
> Indeed, the expectation notation over $D_{\rm cal}$ in Section 3.2 was not intended. We meant the population risk, i.e. the expected loss over a test point $(X, Y)$ as we had in Section B.2. We have modified this in our revision.
>
> > I would suggest reconsidering the use of "exact coverage" in Section 3.3, as this can be conflated for the case where the bound $P(Y \in C(X)) \ge 1-\alpha$ holds with equality.
>
> We see this causing the potential confusion with “exactly $1-\alpha$ coverage” as well. We have modified the “exact coverage” -> “valid coverage” or “at least $1-\alpha$ coverage” in our revision.
>
> We have also fixed the other typos according to your suggestions.

---

> > ### Comment · Reviewer_bDLk · 2021-12-01
> > **Reply to response**
> >
> > Thank you to the authors for their time and effort in replying in detail to the review, and for the updates to the original manuscript! I believe that many of my original concerns have been addressed. My recommendation for acceptance still stands, though like Reviewer Pu7a, my score remains the same (as I don't see any convincing reasons to change it significantly in either direction).

---

### Official Review · Reviewer_G6jH · 2021-11-01

**Correctness:** 4
**Technical Novelty And Significance:** 2
**Empirical Novelty And Significance:** 2
**Recommendation:** 6
**Confidence:** 4

**Main Review:**

**Strengths:**
1. The paper considers an important and interesting issue (i.e., maximizing efficiency of conformal prediction).
2. The proposed approach is evaluated over various setups and datasets (i.e., 9 regression datasets, 7 multi-out regression datasets, and 2 classification datasets).


**Concerns:**
1. I agree that reducing size in conformal prediction is important issue, but the way that this paper considers this issue is currently less motivational; one beauty of (inductive) conformal prediction is that it isolates the issue of choosing the score function and the issue of constructing a conformal predictor---i.e., the guarantee of the constructed conformal prediction holds for any score functions. This implies that we can isolate the task of learning the score function by empirical researchers and the task of conformal predictor construction by theory researchers. Moreover, as we have a better score function (e.g., by devising better network architectures), the size of a conformal predictor is naturally reduced---this can be easily verified if the score function is perfect (e.g., true classifier in classification). However, the proposed approach mixes up the two isolated procedures and making it complex for getting better conformal predictors. What's the problem of designing a better score function to get efficient conformal predictors?

2. Related to above issue, important motivational experiments are missing; for example, in the experiment of Table 1, CQR could be more efficient in terms of interval size by including D_{cal} into D_{train} (i.e., improving the score function by having more training samples); here hyperparameters can be chosen as the paper chooses for \theta. (1) what's the performance of CQR where the quantile regressor is trained over D_{cal} + D_{train} with pinball loss? (2) what's the performance of CQR where the quantile regressor is trained over D_{train} with pinball loss and then the last layer is fine-tuned over D_{cal} with the same pinball loss?

3. for all experiments, I think the variance statistics over random trials would be necessary to show the significance of the proposed approach. Also, the current number of random trials (i.e., 8) is small; I do believe that the paper could be stronger if the proposed approach is evaluated over larger trials (e.g., 100), which is possible if a score function is trained only once with a fixed training set as in inductive conformal prediction. As in the current form, I'm not sure if we can compare lengths among approaches without having the same or very similar coverage---as mentioned in the paper, coverage also affects on the efficiency, so the better efficiency could be due to some randomness of calibration splits.

4. Section 1.1 "Both works formulate this task as a risk minimization problem": I think the earlier work [R1] already consider conformal prediction in the ERM framework, where it also allows any function class and efficiency loss, along with one-parameter special case as their approach; it's better to acknowledge similar prior work and mention differences.

5. "3.2 Theory" is not well connected to the final algorithm; basically the proposed algorithm is (1) fine-tune the score function with a proper efficiency loss and (2) run a known conformal prediction for coverage guarantee. What's the novelty compared to known analyses on the generalization bound?

6. In Table 5, it's better to add comparison results to make the results stronger; the standard conformal prediction can be applicable here.

*[R1]: https://arxiv.org/abs/2001.00106

**Summary Of The Paper:**

This paper considers to improve efficiency of conformal prediction (measured in the "size" of prediction sets). To this end, this paper uses multiple-learnable parameters by generalizing single-parameter conformal prediction. By doing so, the paper demonstrate that the proposed approach can improve the efficiency of conformal predictors while satisfying valid coverage. The efficacy of the proposed approach is demonstrated over regression, multi-output regression, and classification.

**Summary Of The Review:**

This paper considers interesting and important problem, and the proposed approach is broadly evaluated; but as mentioned in the main review, I lean to reject though I'm willing to adjust my understanding and score.



==== POST-REBUTTAL ====

Thanks for the additional discussion and experiments; I think the response mostly addressed my concerns so raise my score to 6. To my understanding, the key message of this paper is that finetune a base predictor with respect to a desired length metric (along with a coverage constraint) if we want to improve the efficiency (in the same length metric) of the final conformal predictor given fixed sample size; this is likely to be true and is more convincing given the additional experiments. As one minor note, [R1] is applicable to both classification and regression as demonstrated in their experiments.

---

> ### Author Response · Authors · 2021-11-22
> **Response to Reviewer G6jH**
>
> Thank you for your thoughtful feedback on our paper! We have performed some additional experiments and revised our paper according to your comments; all modifications other than fixed typos are marked in red for clarity.
>
> We respond to your comments as follows.
>
> > the way that this paper considers this issue is currently less motivational… This implies that we can isolate the task of learning the score function by empirical researchers and the task of conformal predictor construction by theory researchers… What's the problem of designing a better score function to get efficient conformal predictors?
>
> We agree that in vanilla conformal prediction, the score function is isolated from the main conformal step, and could be chosen carefully to additionally optimize an efficiency loss function (such as size). However, we would like to emphasize that there is a lack of principled and unified approach for doing this in the literature, to the best of our knowledge.
>
> Currently, the procedure of designing a better score (such as architecture of base predictor, or form of prediction set) is a non-trivial and “manual” task:
> * often relies on specific domain knowledge about the task at hand;
> * is often done in conjunction with conformal prediction, in multiple rounds of trial-and-error by the researcher;
> * is done by thinking about *high-level properties* of the efficiency loss and coverage constraints (e.g. thinking about what makes the length short), but not by *directly optimizing* the efficiency-coverage trade-off in a *data-dependent* way.
>
> In this sense, we believe our algorithm can be indeed viewed as an automatic, optimized, and data-dependent way for designing the score function, and thus a useful addition to the conformal prediction literature. This was highlighted in our experiments: For example, for the multi-output regression task where we want a minimum-volume prediction set (Section 5.2), it was previously unclear how to choose the shape (ratios between the axis lengths) of the box-shaped predictor manually. In contrast, our CP-Gen optimizes the shape in a data-dependent fashion and outperforms sensible baseline choices of shapes (Section 5.2, Appendix H.3).
>
> > Moreover, as we have a better score function (e.g., by devising better network architectures), the size of a conformal predictor is naturally reduced---this can be easily verified if the score function is perfect (e.g., true classifier in classification). However, the proposed approach mixes up the two isolated procedures and making it complex for getting better conformal predictors.
>
> We agree that for *certain tasks*, the true classifier achieves the best efficiency (e.g. size of label prediction set for classification). However, for other tasks, the most efficient prediction set may not be directly obtained from the (usually conceived) best base predictor. For example, in multi-output regression, having a population best (minimum L2 loss) neural net point prediction function is insufficient for determining the minimum-volume box-shaped predictor, as the variances and correlations come into play. While this could be potentially resolved by studying properties of population best prediction set for the task at hand and approximating the best base predictor with a good neural net, such a procedure has to be done differently for each task and each efficiency loss, and it may be unclear how to do this especially if the desired efficiency loss is unconventional (something else other than size or length).
>
> By contrast, our approach does not aim to mathematically characterize the population best prediction set, but instead optimizes the efficiency loss within any given class $\\{ C_{\theta, t}\\} $. Therefore, we believe our approach is complementary, and could often be more flexible.
>
> Our related work section had some discussions related to this comparison (Section 1.1, paragraph 2 of the “Optimizing efficiency in addition to valid coverage” part).
>
> > important motivational experiments are missing… (1) what's the performance of CQR where the quantile regressor is trained over D_{cal} + D_{train} with pinball loss? (2) what's the performance of CQR where the quantile regressor is trained over D_{train} with pinball loss and then the last layer is fine-tuned over D_{cal} with the same pinball loss?
>
> Thank you for this suggestion. We agree that either having more training data or finetuning with other loss functions (such as the pinball loss) may potentially improve the length for the CQR baseline.
>
> We performed additional experiments to test out these tweaked baselines in Appendix H.2. We find method (1) does improve both the length and the pinball loss slightly, while method (2) is unable to improve either the length or pinball loss (albeit efforts for tuning the optimization). Our CP-Gen-Recal still achieves a better length than both new baselines on most datasets (except for kin8nm).

---

> > ### Author Response · Authors · 2021-11-22
> > **Response to Reviewer G6jH (cont'd)**
> >
> > > variance statistics over random trials would be necessary… the current number of random trials (i.e., 8) is small; I do believe that the paper could be stronger if the proposed approach is evaluated over larger trials (e.g., 100)
> >
> > We tried out 100 random seeds for Table 1 & 2 and reported both the means and stds in Table 14-16. For both experiments, the means did not change much from 8 -> 100 seeds, and the stds indicate significant gains for the multi-output experiments and most of the CQR prediction interval experiments.
> >
> > >  the earlier work [R1] already consider conformal prediction in the ERM framework
> >
> > Thank you for pointing out this related work. This approach could indeed optimize any efficiency loss with any base predictor, though it’s only restricted to classification (and uses classification calibration as a main step). We have properly cited this paper in our revision.
> >
> > > "3.2 Theory" is not well connected to the final algorithm; basically the proposed algorithm is (1) fine-tune the score function with a proper efficiency loss and (2) run a known conformal prediction for coverage guarantee. What's the novelty compared to known analyses on the generalization bound?
> >
> > Indeed, in Section 3.2 we were not claiming novelties in the proof techniques---We mentioned “by standard generalization theory” on Page 5 and “proofs for both results use standard concentration arguments” on Page 6. Rather, the point was to give concrete examples that our CP-Gen algorithm (with the constrained ERM procedure) could enjoy good generalization (in the coverage and efficiency losses) as long as the number of additional trainable parameters (dimension of $\theta$) is small.
> >
> > > In Table 5, it's better to add comparison results to make the results stronger; the standard conformal prediction can be applicable here.
> >
> > In Table 5 we did already use standard conformal prediction as baselines (more concretely the Least-Ambiguous Set-Valued Classifier (LAC) method of Sadinle et al. 2019). To highlight that conformal prediction was used in our baselines too, we have changed the method names to “best *conformalized* single model” and “*conformalized* uniform ensemble” in our revision.
> >
> > Thank you again for your time for reading our response. We would sincerely appreciate it if you could consider raising the scores, if we have addressed your concerns.

---

### Official Review · Reviewer_Pu7a · 2021-11-02

**Correctness:** 4
**Technical Novelty And Significance:** 2
**Empirical Novelty And Significance:** Not applicable
**Recommendation:** 6
**Confidence:** 4

**Main Review:**

## Strengths

The paper is a useful sequel to the earlier work of Yang & Kuchibhotla (2021). Now that more people are aware of conformal methods, it makes sense to develop methods that incorporate tuning of learning algorithms with the goal of obtaining the most precise predictive confidence sets.

## Weaknesses

1. *Clarity*: Definitions or details that are important for understanding the paper are frequently postponed without warning.

- The explicit meaning of *parameter* (as in "a single parameter" or "multiple learnable parameters") is left unspecified till p. 4, where the problem is formally described. At the same time, it seems eminently possible to communicate the content of earlier pages using more intuitive language. For example, consider the following sentence: "Conformal prediction is a powerful technique for learning prediction sets with valid coverage, yet by default its conformalization step only learns a single parameter, and does not optimize the efficiency over more expressive function classes." All this is saying is that the existing approaches treat the learning algorithm as completely fixed (so that conditional on the observed data, there is only one set prediction map for each confidence level), whereas the proposed methods aim to find the most precise set prediction map over a *collection* of learning algorithms, where the collection is allowed to be uncountably infinite.

- Similarly, the question of where the extra parameter $\theta$ is coming from or how it may arise is not addressed till p. 8. For example, Figure 1 has less impact because it cannot be clear to the reader at the time what $\theta$ represents in the context of CQR. As a result, the paper reads mainly like a purely theoretical exercise until concrete instances of $\theta$ are finally given in the last sections. (This is just a suggestion, but consider starting the paper with a concrete example from Section 5 that is well-defined.)

- The descriptions of the experimental setups ought to be more clear, too. For example, it is doubtful how much of Section 5.1 can be understood *without* Romano et al. (2019). The symbols $\hat F$, $\hat f_{\text{lo}}$, $\hat f_{\text{hi}}$, etc. do not appear to be defined anywhere, and the symbol $d_h$ is defined 7 lines after its first occurrence. I also do not understand what the authors mean by "1r decay."

- I am confused about the experiment in Section 5.2. In light of the stated motivation for the proposed method, I would have thought that the interesting comparisons are the ones in which first, a parametric family of learning algorithms is fixed, and then the method that does this for a fixed $\theta_0$ is compared to the proposed method that learns an optimal $\hat \theta$. It is not clear to me if the two baseline methods are comparable in this manner. (By the way, a slightly less naive method that adapts to the correlation structure would probably utilize the maximum statistic, e.g., $\|Y - \hat f(X)\|_\infty$, rather than a Bonferroni bound.)

2. *Balance / Completeness*: The paper actually proposes *two* meta-algorithms: CP-Gen and CP-Gen-Recal. The theory sections are almost exclusively concerned with CP-Gen (with the exception of Section 3.3), whereas the experimental results are only reported for CP-Gen-Recal. This has created an odd imbalance in the manuscript. I feel like either the experimental results for CP-Gen ought to be included in the paper as well as the content of Appendix G.3, or most of the material pertaining to CP-Gen ought to be moved to the appendix.

3. *Originality*: As of writing, I am still trying to decide whether the contributions of this paper are substantial enough in light of the contributions of earlier works. From the point of view of theoretical validity, many of the ideas here appear to be already present in, say, Yang & Kuchibhotla (2021), although it is true that they have not been worked out at this level of detail. On the other hand, from the point of view of practical implementation, optimizing over an arbitrary and large $\Theta$ is a nontrivial challenge, and the authors do well to borrow from a different line of uncertainty quantification literature. This may well be a matter of emphasis or branding. In any case, I would like to hear other opinions on this point.

## Typos
- (p. 5. The line above Section 3.2) $\epsilon = 0$ -> $\epsilon_0 = 0$.
- (p. 9. The line after the itemized list) Our datasets is -> Our datasets are
- (p. 9. **Additional experiment**) The citation for the ImageNet dataset is incorrectly given as (Angelopoulos et al., 2020).
- (Section B.2 from after Eq. (11)) $\hat L_{\text{coverage}}$  -> $\hat L_{\text{eff}}$ and $L_{\text{coverage}}$  -> $L_{\text{eff}}$ throughout
- (Section B.3) $R_n(\mathcal{C})$ is used, but $R^{\text{eff}}_{n{\text{cal}}}(\mathcal{C})$ was given on p. 6.
- (p. 16, Paragraph 4) a shortest prediction interval -> the shortest prediction interval
- (p. 17. The last unnumbered display equation) Remove , from the subscript.
- (p. 18. The line after the first unnumbered display equation) Add "by" between "and" and "the assumption."
- (p. 18) $F_{\hat \theta}$, $\hat F^{\text{cal}}_{\hat \theta}$, etc. all look too similar, making the proof unnecessarily difficult to read.
- (p. 21) On p. 3, $K$ was used to denote the number of classes. Here, $L$ is used to denote the number of classes, and $K$ now represents the number of base predictors in an ensemble.
- (p. 21) $p$ in the exponent can be confused with $p_\theta$.
- (p. 21) In **Methods for learning prediction sets**, $i$ denotes the epoch count. However, $i$ is already being used to denote a single base predictor in an ensemble.

**Summary Of The Paper:**

This paper proposes conformal methods for learning a predictive confidence set that is guaranteed to be nearly the most statistically efficient among predictive confidence sets produced by a parametric class of learning algorithms. To explain, many wrapper methods for assessing prediction uncertainty assume that the user has already committed to one particular learning algorithm and aim to quantify uncertainty in predictions produced by using *this* learning algorithm. By contrast, the methods proposed here assume that the learning algorithm has been specified only up to some parameter $\theta$ so that it makes sense to optimize $\theta$ with respect to the precision of the resulting predictive confidence sets.

From the point of view of practical implementation, the obvious challenge is the optimization of $\theta$ (as well as the associated prediction sets) over an arbitrary and large space of possible values of $\theta$, as the optimization involves a hard constraint. Thus, one main contribution of the paper is to propose a differentiable proxy that allows an efficient search over the parameter space. Regarding theoretical guarantees, the paper proves that the original, non-differentiable formulation of one of the methods achieves approximately valid coverage and nearly optimal statistical efficiency when the class of learning algorithms has a low capacity.

**Summary Of The Review:**

Overall, I think the paper is a useful extension of some earlier works on uncertainty quantification with an eye towards statistical efficiency. I tentatively recommend it for acceptance with some reservations.

---

> ### Author Response · Authors · 2021-11-22
> **Response to Reviewer Pu7a**
>
> Thank you for your positive feedback and very detailed and thoughtful comments on our paper! We have updated our paper according to your various suggestions on the presentation and clarity; all modifications other than fixed typos are marked in red for clarity.
>
> We respond to your specific concerns as follows.
>
> > Explicit meaning of “parameter”... Figure 1 has less impact… unclear what $\theta$ represents in the context of CQR… consider starting the paper with a concrete example
>
> We agree the meaning of “parameter” may have been unclear to the reader. We changed our writing accordingly in our revision: (1) In the Introduction, bottom of Page 1, we added a new sentence talking about how CQR only optimizes 1 parameter. (2) We have expanded the caption in Figure 1 to explain what $\theta$ means (last linear layer within the quantile network), and added a pointer of Section 5.1 (along with original pointer of Appendix E.3).
>
> > How much of Section 5.1 can be understood without Romano et al. (2019)... symbols do not appear to be defined… symbol $d_h$... “1r decay”
>
> We expanded the descriptions within Section 5.1 to clarify what $\widehat{f}_{\rm lo}, \widehat{f}_{\rm hi}$, and $d_h$ means. We also added the formula of the pinball losses (used to train these quantile functions) in Appendix E.1. The “1r decay” was actually “lr decay” by which we meant learning rate decay; we have modified it to “learning rate decay” to avoid confusion.
>
> > experiment in Section 5.2… interesting comparisons are the ones in which first, a parametric family of learning algorithms is fixed, and then the method that does this for a fixed $\theta_0$ is compared to the proposed method that learns an optimal $\widehat{\theta}$... a slightly less naive method that adapts to the correlation structure would probably utilize the maximum statistic, e.g., $|Y−\widehat{f}(X)|_\infty$, rather than a Bonferroni bound.
>
> Thank you for this insightful question. We agree that $|Y−\widehat{f}(X)|_\infty$ is a good score function that gives a baseline conformal method. This score function would yield a hyper-*cube* shaped predictor, rather than a hyper-rectangle shaped predictor. Also, this method does correspond to fixing a particular $\theta_0$ (recall here our parametrization is that $\theta$ represents the ratios between $u_i$’s, so this baseline is just taking $\theta_0$ being the *all-ones* ratio).
>
> Experimentally, we tested this new Max-score-Conformal baseline method and reported the results in Appendix H.3. We found this baseline improves the volume over the original Coord-wise-Recal baseline on one particular dataset (Cartpole), but significantly worse volumes on many other datasets (Ant, Walker, Humanoid). For the latter we found that it is due to the max score pays a lot in terms of the volume, as the hypercube shaped predictor is unnecessarily large when most of the states are easy to predict (small conditional variance) but a few states are hard to predict (so that size of the hypercube is determined by those dimensions). Our CP-Gen-Recal still achieves better volumes than both baselines on all datasets.
>
> We also remark that our original Coord-wise-Recal baseline also corresponds to using a fixed $\theta_0$, which is the ratios obtained by coordinate wise conformal prediction. We have clarified this point in our revision in Appendix H.3.
>
> > Balance/Completeness… theory sections are almost exclusively concerned with CP-Gen (with the exception of Section 3.3), whereas the experimental results are only reported for CP-Gen-Recal. This has created an odd imbalance in the manuscript.
>
> We agree that this may have created a somewhat mismatch between the theory and empirical part. This design was kind of intended though---We thought CP-Gen has a more informative theory, whereas CP-Gen-Recal is a better empirical algorithm (non-approximate valid coverage is often desired in practice). Therefore in the main text we have focused the theory on CP-Gen and the experiments on CP-Gen-Recal; and put most of the materials for the other way (theory for CP-Gen-Recal, experiments for CP-Gen) into the appendix.
>
> To address the reader’s potential confusion about why the theory part is more about CP-Gen, we have added a few sentences at the beginning of Section 3.2 to explain our reasoning.

---

> > ### Author Response · Authors · 2021-11-22
> > **Response to Reviewer Pu7a (cont'd)**
> >
> > **Typos**
> >
> > Thank you for pointing out the typos; we have fixed them accordingly in our revision. Here are some minor comments about our fixes:
> >
> > > (p. 9. Additional experiment) The citation for the ImageNet dataset is incorrectly given as (Angelopoulos et al., 2020)
> >
> > The citation of (Angelopoulos et al., 2020) was originally for their label prediction set experiment on ImageNet, not the ImageNet dataset. To avoid confusion we removed this citation; both the ImageNet paper (Deng et al. 2009) and (Angelopoulos et al. 2020) are now still cited at the beginning of Appendix F.
> >
> > > (p. 21) On p. 3, K was used to denote the number of classes. Here, L is used to denote the number of classes… $p$ in the exponent can be confused with $p_\theta$... In Methods for learning prediction sets, $i$ denotes the epoch count. However, $i$ is already being used to denote a single base predictor in an ensemble…
> >
> > We have modified all these notation accordingly: $K$ -> $L$ for number of classes on Page 3; $p$ in the exponent -> $q$ ($L_q^q$ norm), and epoch index $i$ -> $j$.

---

> > > ### Comment · Reviewer_Pu7a · 2021-11-30
> > > **Some further suggestions**
> > >
> > > I thank the authors for taking their valuable time to consider and respond to my comments. The revision addressed many of my concerns about clarity, and in this respect, the reservations I had about recommending this paper have been removed. I am keeping my score at 6, as I did not see anything in the other reviews or responses that made me want to change my final score significantly in either direction.
> > >
> > > Here are some additional suggestions. (These would have no impact on my final score.)
> > > 1. I found the authors' first two comments to Reviewer G6hH about the paper's motivation quite illuminating. While it looks like most of what the authors say in those two comments already appears in the paper, I feel like the ideas are more clearly expressed in those two comments than they are in the paper. Perhaps it is worth considering editing relevant parts of the text to be more like these two comments.
> > > 2. There is still something that does not sit quite well about Section 3.2. I noticed that Reviewers G6hH and bDLk also had comments about the same section; I think we are all speaking from the same uneasiness. Section 3.2 feels disconnected from the rest of the paper because (1) it is not directly about the algorithm that is actually used in practice, and (2) the proof techniques involved are fairly standard. This is an issue because the section takes up a significant amount of space and is singled out as the place where one of the main contributions is being made. Perhaps what is needed is (1) a more persuasive argument for _why_ the theory without reconformalization is more informative, as well as (2) a clarification about what, if anything, these results tell us about the performance of the reconformalized algorithm.
> > > 3. When reporting experimental results, please consider placing numbers that are meant to be compared side by side in close proximity. For example, in Table 1, we want to compare Coverage / Length / $L^{\text{test}}_{\text{pinball}}$ for CQR vs QR + CP-Gen-Recal for the same dataset. Thus, it would make things easier for the reader to have the coverage numbers side-by-side, etc. The same comment applies to all the other tables.

---

### Author Response · Authors · 2021-11-22
**Revision Uploaded**

We thank all reviewers again for their very detailed and thoughtful feedback on our paper. We have revised our paper accordingly to incorporate all reviewers' suggestions. All changes except for fixed typoes are marked in red, for clarity.

Notably, Appendix H in our revision now includes several additional experiments and analyses as suggested by the reviewers.

* Analyses of conditional coverage of CP-Gen vs. CQR (Appendix H.1);
* Alternative tweaked versions of the CQR baseline (Appendix H.2);
* An additional Max-score-Conformal baseline for the multi-output regression experiment (Appendix H.3);
* Results for Table 1, 2 with 100 random seeds, reporting means and standard deviations of the runs (Appendix H.4).

We also briefly re-highlight our **contributions in the multi-output regression experiment** in Section 5.2, which captures new challenges different from our CQR experiment. This task requires finding a hyperrectangle (box)-shaped predictor that covers the true multi-dimensional output with at least 90% coverage probability, and small volume. While it is easy to *proportionally scale* any box-shaped predictor to achieve valid coverage by standard conformal prediction, it is harder to determine the *shape* of the predictor (i.e. ratios between the axis lengths) that achieves the minimum volume.

As we showed in Section 5.2 & Appendix H.3, both coordinate-wise conformal prediction and the all ones ratio (i.e. hyper*cube*-shaped predictor) can be scaled to achieve valid coverage, but they do not always yield a good volume. Instead, our CP-Gen-Recal explicitly optimizes the ratios on the constrained ERM problem and obtains significantly better volumes than the above two baselines.

We believe this experiment showcases our CP-Gen algorithm as an automatic and data-dependent way for optimizing any efficiency loss subject to valid coverage, which is especially desirable when the efficiency loss is unconventional (something else other than size or length).

---

> ### Public Comment · ~Yu_Bai1 · 2022-03-22
> **Github code**
>
> Our open source code is now released at https://github.com/allenbai01/cp-gen.

---

### Decision · Program_Chairs · 2022-01-20

**Decision:**

Accept (Poster)

**Comment:**

This paper describes a few practically relevant extensions of the conformal prediction framework, that has recently become popular in the ML community for providing (marginally valid) prediction sets without making distributional assumptions. The conceptual contributions are not major, given existing work --- without recalibration, the main idea of optimizing over two parameters was explored by Yang and Kuchibhotla (and is well understood even before YK, albeit not fleshed out). The current paper generalizes YK, and with the additional recalibration dataset, it is again simply an instance of standard conformal prediction. The optimization via Lagrangians is a nice addition, but it is ultimately a heuristic that performs well in practice. Nevertheless, the paper is well written, and the experiments are well done, making this a good contribution for practitioners. I recommend acceptance, and congratulate the authors on a nice work.

As a minor note, Remark 1 should not be attributed to [AB21], since it is a well known fact and deserves an earlier reference.